# Button shear testing for adhesion measurements of 2D materials

Josef Schätz [1,2], Navin Nayi[1], Jonas Weber [3,4], Christoph Metzke [3,5], Sebastian Lukas [2], Jürgen Walter[1], Tim Schaffus[1], Fabian Streb [1], Eros Reato[2], Agata Piacentini[2,6], Annika Grundmann[7], Holger Kalisch[7], Michael Heuken[7,8], Andrei Vescan[7], Stephan Pindl[1] & Max C. Lemme [2,6] ✉

Two-dimensional (2D) materials are considered for numerous applications in microelectronics, although several challenges remain when integrating them into functional devices. Weak adhesion is one of them, caused by their chemical inertness. Quantifying the adhesion of 2D materials on three-dimensional surfaces is, therefore, an essential step toward reliable 2D device integration. To this end, button shear testing is proposed and demonstrated as a method for evaluating the adhesion of 2D materials with the examples of graphene, hexagonal boron nitride (hBN), molybdenum disulfide, and tungsten diselenide on silicon dioxide and silicon nitride substrates. We propose a fabrication process flow for polymer buttons on the 2D materials and establish suitable button dimensions and testing shear speeds. We show with our quantitative data that low substrate roughness and oxygen plasma treatments on the substrates before 2D material transfer result in higher shear strengths. Thermal annealing increases the adhesion of hBN on silicon dioxide and correlates with the thermal interface resistance between these materials. This establishes button shear testing as a reliable and repeatable method for quantifying the adhesion of 2D materials.

The integration of two-dimensional (2D) materials into semiconductor devices is gaining momentum as the technology matures[1–3]. One of the remaining challenges is the adhesion, or lack thereof, of the 2D films to adjacent layers due to their van der Waals (vdW) nature[4,5]. Adhesion ultimately affects the device reliability, but also 2D layer transfer methods[6,7]. Although there are approaches to generally improve adhesion[8,9], a quantitative and reliable method for measuring and evaluating the adhesion of 2D materials is lacking. Existing methodologies for measuring absolute adhesion energy values like blister tests[10–16], nanoparticles[17–19], and atomic force microscopy (AFM)[20–26] are predominantly rather time-consuming. A shearing technique for interlayer cleavage energy of graphite was proposed by Wang et al.[27]. Other methods like substrate streching[28–32] are restricted to specific substrates. The double cantilever beam method[33–35] is applicable to gather information on the average adhesion of large area interfaces in the millimeter range but cannot be applied for local measurements in the micrometer range. A promising method for some applications is scratch testing[36–38]. However, the mechanical properties of all

[1]Infineon Technologies AG, Wernerwerkstraße 2, 93049 Regensburg, Germany. [2]Chair of Electronic Devices, RWTH Aachen University, Otto-Blumenthal-Str. 25, 52074 Aachen, Germany. [3]Department of Electrical Engineering and Media Technology, Deggendorf Institute of Technology, Dieter-Görlitz-Platz 1, 94469 Deggendorf, Germany. [4]Department of Applied Physics, University of Barcelona, Martí i Franquès 1, 08028 Barcelona, Spain. [5]Department of Electrical Engineering, Helmut Schmidt University/University of the Federal Armed Forces Hamburg, Holstenhofweg 85, 22043 Hamburg, Germany. [6]AMO GmbH, Advanced Microelectronic Center Aachen, Otto-Blumenthal-Str. 25, 52074 Aachen, Germany. [7]Compound Semiconductor Technology, RWTH Aachen University, Sommerfeldstr. 18, 52074 Aachen, Germany. [8]AIXTRON SE, Dornkaulstr. 2, 52134 Herzogenrath, Germany. ✉e-mail: max.lemme@rwth-aachen.de

materials in a sample stack, such as hardness, influence the critical load in a scratch test[39]. This limits the comparability of multilayer systems or of measurements on different substrates. Four-point bending as an established adhesion measurement method[40,41] was recently used to assess the adhesion of transition metal dichalcogenides (TMDC)[42]. All methods can be categorized by their ratio of forces perpendicular (mode I) and parallel (mode II) to the surface, or a mix.

Here, we introduce button shear testing as a quantitative method for determining the shear strengths of 2D materials. Button shear testing is an established method for adhesion measurements in typical semiconductor technologies and materials[43–48], which yields rapid and conclusive results based on vast existing knowledge. We demonstrate the feasibility of the method for different 2D materials and substrate combinations and its application in evaluating the effect of sample treatments on adhesion.

## Results

### Button shear testing

The graphene samples were based on commercial chemical vapor deposited (CVD) monolayer graphene on copper. This was transferred onto three different substrates by a wet-etching process[49] before button fabrication, (a) 250 nm silicon dioxide (SiO$_2$) deposited with oxygen and tetraethyl orthosilicate precursors (TEOS SiO$_2$), (b) 250 nm silicon nitride deposited from ammonia and dichlorosilane (Si$_3$N$_4$), and (c) 90 nm silicon dioxide grown by thermal oxidation of silicon (thermal SiO$_2$). A subset of TEOS SiO$_2$ and thermal SiO$_2$ substrates was treated with oxygen (O$_2$) plasma before graphene transfer. CVD hexagonal boron nitride (hBN) was also transferred from a copper growth substrate onto O$_2$ plasma-treated thermal SiO$_2$ substrates by a wet-etching process[49], and subsets with hBN were annealed up to 1000 °C between the transfer process and button fabrication. Metal-organic

chemical-vapor-deposited (MOCVD) molybdenum disulfide (MoS$_2$)[50,51] and MOCVD tungsten diselenide (WSe$_2$)[52] was transferred from the sapphire growth substrate onto O$_2$ plasma-treated thermal SiO$_2$ substrates. Schematic cross-sections of all samples are shown in Supplementary Fig. 1.

Buttons were fabricated by spin-coating polymethyl methacrylate (PMMA) onto the samples to create a 5 µm PMMA film. A 20 nm thick aluminum hard mask deposition followed by a lithography process step led to structures (buttons) with typical lateral dimensions of 60 ×100 µm. Anisotropic dry etching of the PMMA film in an oxygen (O$_2$) plasma produced the desired buttons (Fig. 1a, b). Polystyrene as an alternative button material did not lead to a cuboid button cross-section after O$_2$ plasma-etching and was excluded from the shear experiments (Supplementary Fig. 2).

Button shear test measurements were performed with a DAGE4000Plus pull-shear tester. The stage with the sample moved with a defined speed to create contact between the fixed shear head and the button (Fig. 1c). The lateral displacement of the stage was then recorded while the force that acts on the shear head was measured by the cartridge. The force increased as the shear head made contact with the buttons. The force reached a maximum at a certain point where the shearing of the button started, moving it slightly on the target (see Fig. 1b). After the initiation of the shear process, less force was required to maintain the shear process, and the measurement was stopped. The maximum recorded force corresponds to the force that is required to initiate the shear process. This is expected to occur at the weakest spot of the interface underneath the button and is labeled as critical shear force $F_C$ within this work.

Reference button shear tests were performed on samples with PMMA on thermal SiO$_2$ and button lengths of 20, 40, and 60 µm. Typically button shear tests use button sizes in the millimeter range

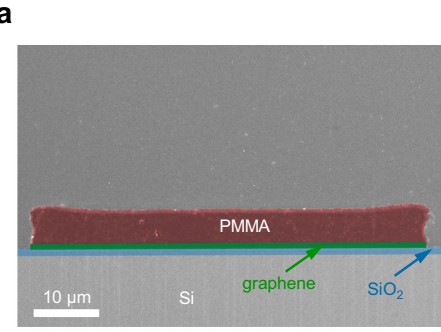

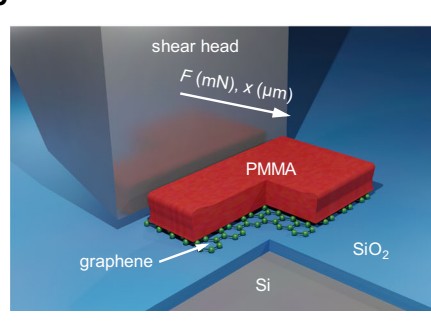

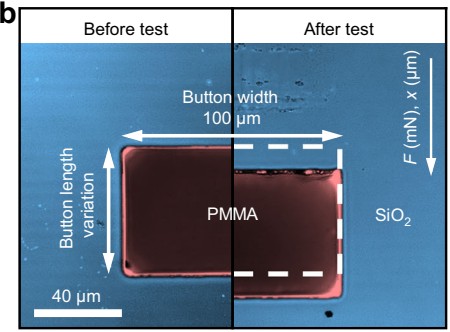

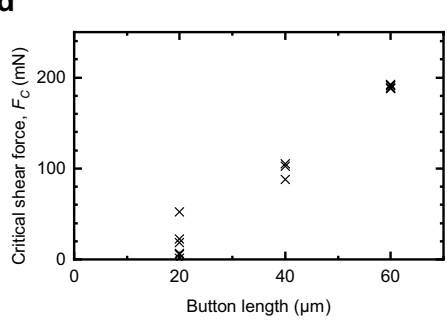

**Fig. 1 | Illustration of button geometry and shear test mechanism. a** Colored scanning electron microscopy cross-section of a fabricated polymethyl methacrylate (PMMA) button (red). SiO$_2$ (blue) and graphene (green) are indicated. **b** Colored microscope image of a PMMA button (red) on SiO$_2$ (blue) before (left) and after (right) button shear testing. The shifted position due to the shear process is visible and the original button position before shear testing is indicated by white dashed lines on the right side. Button length and button width are assigned to the button dimensions. The white arrow on the upper right corner indicates the direction of the force $F$ acting onto the button and the direction of the relative movement along $x$ between shear head and button. **c** Schematic of the button shear test with graphene at the interface of the PMMA button and a SiO$_2$ substrate. The shear head approaches the button along $x$ and will measure $F$. **d** Influence of button length on the critical shear force $F_C$. Measurements were performed with PMMA on SiO$_2$ without 2D material and a fixed button width of 100 µm. Each data point represents a measurement on a separate button. A reliable result can be achieved at 60 µm button length.

and measure $F_C$ in the Newton range[43]. However, 2D materials today must be expected to have cracks, holes, or other defects[53–55] that influence their mechanical properties[32,56,57] in such millimeter-range contact areas. We therefore implemented a measurement routine based on smaller buttons to overcome this challenge. A button length of 60 µm was established as a minimum for reproducible results by these initial experiments because smaller buttons with 20 µm and 40 µm button length show very low and scattered $F_C$ (Fig. 1d). PMMA is a soft material with low elastic modulus compared to other assistance layers in shear testing experiments with 2D materials[58]. That leads to button deformation near the contacted button edge. Hence, the ratio of forces perpendicular (mode I) and parallel (mode II) to the surface varies along the shear path[43,59]. The pronounced mixed mode near the contacted edge can be the reason for the very low $F_C$ at small button lengths[60]. The influence of plasma on PMMA edges[61,62] is an additional possible reason for the $F_C$ variability with smaller button lengths.

Shear tests were further performed on a calibration sample that consists of a step in the (100) silicon surface. This step can be considered as a button that cannot be sheared off to assess the accuracy of the force and displacement measurement. Ideally, the measured force before contacting the stable obstacle is zero and jumps to a very high value upon contact. Here, we chose a shear speed of 1 µm s⁻¹ to be able to stop the measurement in time and prevent damage to the tool. A large cartridge with high internal stiffness and a small cartridge for small forces were compared (Fig. 2a). The large cartridge delivered correct displacement values, but the noise in the force data before contact made reliable force measurements impossible. The small cartridge produced no noise in the force data and was chosen for the experiments. However, the pronounced force-dependent stiffness of the small cartridge leads to non-reliable displacement measurements. A multipoint calibration may reduce this effect, but there are several other system-dependent influences on the displacement measurement[45]. Therefore, no quantitative evaluation of the displacement was performed, but $F_C$ was extracted only from the force data. Dividing $F_C$ by the button area 100 µm x 60 µm leads to the area-independent shear strength $\tau_C$.

The influence of the shear speed was evaluated on samples with PMMA on thermal SiO₂ because shear speed effects strongly influence $\tau_C$[31,35,48]. $\tau_C$ at low shear speeds below 5 µm s⁻¹ was significantly lower than at high shear speeds (Fig. 2b, c). Similar trends were observed in previous studies[48,63–66] and can be assigned to viscoelastic properties of PMMA[67–70]. We chose a shear speed of 10 µm s⁻¹ for the experiments with 2D materials to prevent a significant influence of the viscoelastic properties of PMMA.

## Graphene

The shear strength of graphene on the three substrates TEOS SiO₂, Si₃N₄, and thermal SiO₂ was measured with the optimized parameters of a small cartridge and 10 µm s⁻¹ shear speed. Figure 3a compares $\tau_C$ of graphene on as deposited TEOS SiO₂ (1.55 ± 0.31 MPa), Si₃N₄ (2.88 ± 0.31 MPa), and thermal SiO₂ (2.68 ± 0.11 MPa). We correlated these results with the surface roughness $s_a$ of the substrates and the surface roughness of graphene on the substrates as the roughness is known to influence the adhesion of a 2D material[21,71]. AFM roughness data show that bare TEOS SiO₂ has a roughness of $s_a = 0.76$ nm and graphene on TEOS SiO₂ has a lower roughness of $s_a = 0.64$ nm. On the smoother substrate thermal SiO₂ the roughness remains at $s_a = 0.22$ nm without and with graphene. On Si₃N₄ the roughness increases from $s_a = 0.30$ nm without graphene to $s_a = 0.42$ nm with graphene (see Fig. 3b–d and Table 1). This increase in roughness may be attributed to PMMA residues from the wet etching transfer process[72,73] and limits the validity of roughness measurements on transferred CVD-grown graphene. However, the lower roughness of graphene on TEOS SiO₂ compared to bare TEOS SiO₂ is an indication that the graphene does not fully follow the TEOS SiO₂ morphology. This behavior on rough substrates is predicted in theoretical studies[74,75] and leads to a smaller effective interface area and finally to a smaller adhesion[76].

The TEOS SiO₂ and thermal SiO₂ samples were further exposed to O₂ plasma. This was not the case for the Si₃N₄ sample, because an O₂ plasma would have modified the surface to an undefined silicon-oxide-nitride composition[8]. The O₂ plasma treatment increased $\tau_C$ of graphene on TEOS SiO₂ to 2.21 ± 0.33 MPa, but we did not observe a significant change on thermal SiO₂ (2.77 ± 0.32 MPa, Fig. 3a). AFM measurements show that the roughness of our oxides did not change significantly after the O₂ plasma, and, hence, this was ruled out as the reason for the different behavior (Table 1). However, several other potential explanations for the influence of the plasma treatment on adhesion remain: O₂ plasma reduces the water (H₂O) contact angle ($CA_{water}$), increases the surface energy[8,21,23], and reduces carbon contamination on the surface[8]. Contact angle measurements showed that the applied O₂ plasma reduced the $CA_{water}$ from 7.3° to 3.0° for TEOS SiO₂ and from 42.7° to 2.7° for thermal SiO₂, (Fig. 3e-g and Table 1). Since the reduction of $CA_{water}$ is more pronounced on thermal SiO₂, which showed no significant change in $\tau_C$, we rule this effect out as the main contributor. X-ray photoelectron spectroscopy (XPS) measurements of C 1 $s$ peak on TEOS SiO₂ (Fig. 3h) and thermal SiO₂ (Fig. 3i) before and after O₂ plasma show that the reduction of carbon contamination on the surface is more pronounced on TEOS

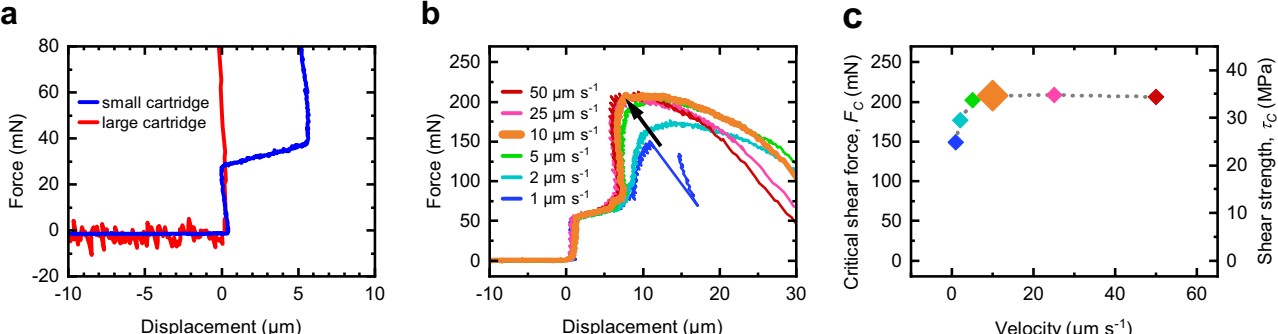

**Fig. 2 | Characterization of cartridge and influence of shear speed. a** Force vs. displacement curves of measurements with two different cartridges contacting a stable obstacle. The small cartridge (blue) delivers a lower noise in the force signal at the cost of no reliable displacement data compared to the large cartridge (red). **b** Force vs. displacement curves and **c** extracted critical shear force $F_C$ and shear strength $\tau_C$ on samples with polymethyl methacrylate (PMMA) on SiO₂ at different shear velocities from 1 µm s⁻¹ (blue) to 50 µm s⁻¹ (red). The black arrow in b indicates the increase in $F_C$ and reduction of the displacement at $F_C$ at higher shear velocities. The dotted gray line in c is a guide to the eye connecting the data points. The velocity 10 µm s⁻¹ (orange) is highlighted and was used in further experiments with 2D materials.

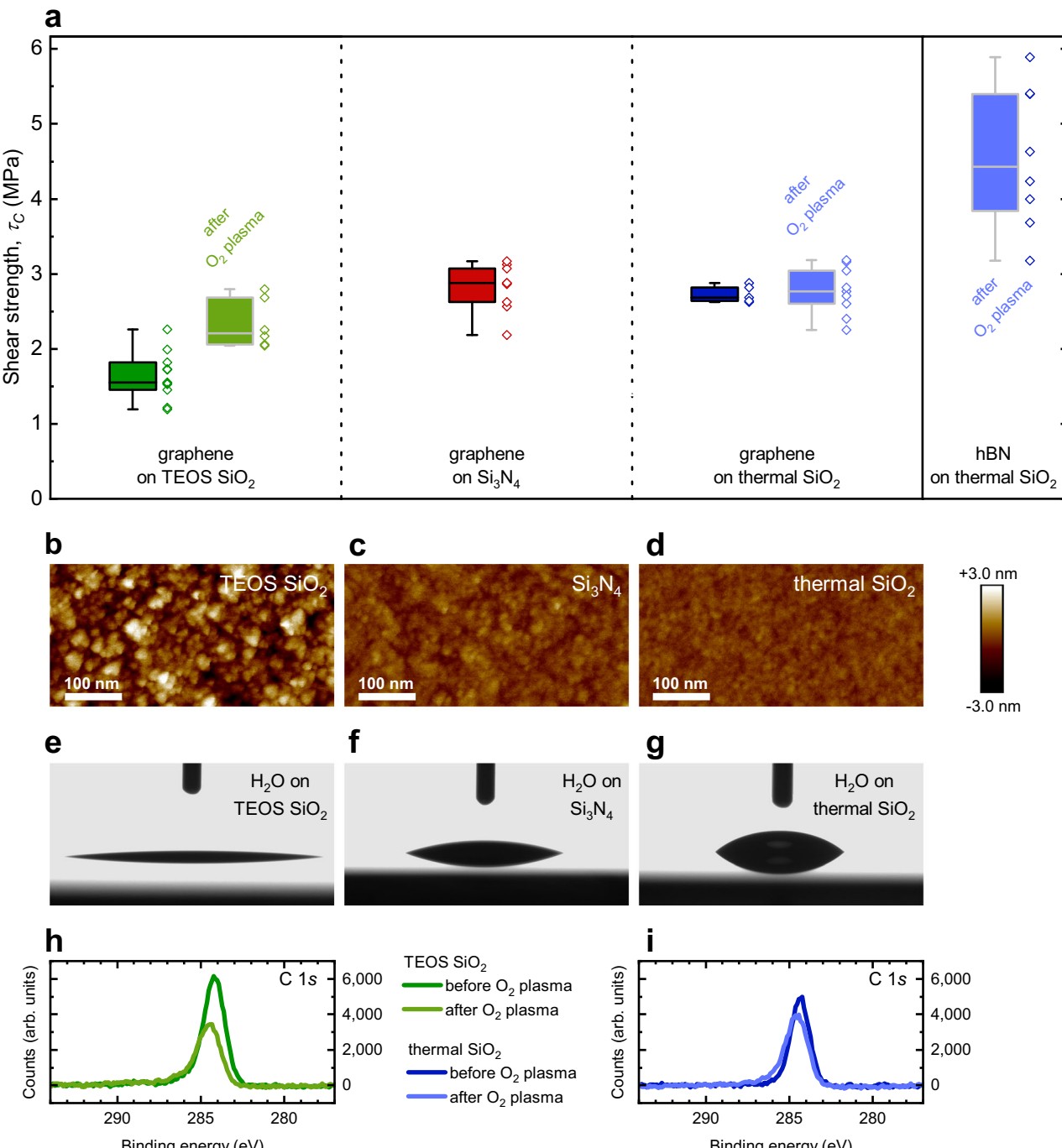

**Fig. 3 | Characterization of tetraethyl orthosilicate (TEOS) SiO₂, Si₃N₄, and thermal SiO₂ substrates and shear strength determination of graphene and hBN on the substrates. a** Shear strength $\tau_C$ of graphene on TEOS SiO₂ (dark green, number of buttons n = 11), Si₃N₄ (red, n = 9), and thermal SiO₂ (dark blue, n = 5). The right blue box plot represents the $\tau_C$ of hBN on thermal SiO₂ (n = 8). Box plots in light green (n = 6) and light blue (n = 9) show measurements of samples with O₂ plasma treatment before the transfer of 2D material. Box plots indicate median (middle line), 25th, 75th percentile (box) and minimum and maximum (whiskers). **b–d** Atomic force microscopy (AFM) roughness measurements of TEOS SiO₂ ($s_a$ = 0.76 nm), Si₃N₄ ($s_a$ = 0.30 nm), and thermal SiO₂ ($s_a$ = 0.22 nm) without O₂ plasma, respectively. **e-g** Contact angle $CA$ measurements with water on TEOS SiO₂ ($CA_{water}$ = 7.3 ± 1.5 °), Si₃N₄ ($CA_{water}$ = 19.3 ± 0.3 °), and thermal SiO₂ ($CA_{water}$ = 42.7 ± 0.3 °) without O₂ plasma, respectively. **h** X-ray photoelectron spectroscopy (XPS) measurement of C 1$s$ peak on TEOS SiO₂ before (dark green) and after (light green) O₂ plasma treatment. **i** XPS measurement of C 1$s$ peak on thermal SiO₂ before (dark blue) and after (light blue) O₂ plasma treatment.

SiO₂, which confirms previous investigations[8]. Therefore, we conclude that the adhesion increase is dominated by the effect of surface cleaning. We note that only buttons with visibly intact graphene underneath were included in the assessment. This is because the high $CA_{water}$ caused cracking and partial delamination of graphene during the drying process in some cases. Buttons on areas with partially delaminated graphene were excluded to evaluate only pure graphene-substrate-interfaces (see also Supplementary Fig. 3). In all cases, delamination at the substrate-graphene interface occurred. This was confirmed by optical microscopy (Supplementary Fig. 4) and Raman measurements (Supplementary Fig. 5). Minor graphene residues are found predominantly near the contacted button edge. The pronounced mixed mode at the contacted button edge is a likely reason for this observation[60].

**Table 1 | Comparison of contact angle *CA* measurements, roughness $s_a$ measurements, and shear strengths $\tau_C$ of graphene on TEOS SiO$_2$, Si$_3$N$_4$, and thermal SiO$_2$ and of hBN on thermal SiO$_2$**

| | TEOS SiO$_2$ | | Si$_3$N$_4$ | Thermal SiO$_2$ | |
|---|---|---|---|---|---|
| O$_2$ plasma | No | Yes | No | No | Yes |
| $CA_{water}$ (°) | 7.3 ± 1.5 | 3.0 ± 1.3 | 19.3 ± 0.3 | 42.7 ± 0.3 | 2.7 ± 1.6 |
| $CA_{diiodomethane}$ (°) | 40.6 ± 0.4 | 35.3 ± 0.6 | 40.7 ± 1.8 | 45.2 ± 0.5 | 34.6 ± 0.3 |
| Surface Energy (mN m$^{-1}$) | 76.0 ± 0.6 | 77.4 ± 1.9 | 73.0 ± 0.6 | 60.6 ± 0.2 | 77.5 ± 1.8 |
| Roughness $s_a$ of the substrate (nm) | 0.76 | 0.69 | 0.30 | 0.22 | 0.21 |
| Roughness $s_a$ of graphene on the substrate (nm) | 0.64 | | 0.42 | 0.22 | |
| Shear strength $\tau_C$ of graphene (MPa) | 1.55 ± 0.31 | 2.21 ± 0.33 | 2.88 ± 0.31 | 2.68 ± 0.11 | 2.77 ± 0.32 |
| Shear strength $\tau_C$ of hBN (MPa) | | | | | 4.43 ± 0.95 |

## Hexagonal boron nitride

hBN is of interest as a 2D dielectric for encapsulating electrically active 2D materials like graphene or semiconducting TMDCs[1,77,78]. In many integration schemes, hBN is therefore in contact with a three-dimensional substrate like SiO$_2$. We have investigated the adhesion of hBN on thermal SiO$_2$ with O$_2$ plasma cleaning. The $\tau_C$ of 4.43 ± 0.95 MPa is significantly higher on this substrate compared to graphene, which was 2.77 ± 0.32 MPa, but also shows a higher standard deviation (Fig. 3a). This is in contrast with the work by Rokni and Lu[25], who compared the adhesion of graphene and hBN on a silicon oxide substrate and measured a larger interfacial adhesion energy in the case of graphene. However, the high adhesion energy was only apparent when the graphene was pressed onto the substrate by a pressure of at least 3 MPa before measurement, and their experiments were conducted with (nearly) defect-free exfoliated 2D materials. They hypothesized that the large adhesion energy values for graphene on silicon oxide arose from non-vdW forces, i.e. hydrogen bonds (e.g. C-H...O-Si) or maybe even covalent bonds (e.g. C-O-Si)[25]. In our case, we expect a higher defect density of the CVD-grown hBN compared to the CVD-grown graphene because the growth processes are less mature[53]. As an indication of the defect density of graphene and hBN used in this work, we provide laser scanning microscope images, AFM measurements, and Raman measurements in Supplementary Fig. 6. A higher defect density can drastically increase the adhesion, e.g. by hydrogen bonds as shown for graphene on polymers[30,32]. This finding could be used to engineer 2D materials with defects at well-defined locations as a reasonable integration scheme to achieve sufficient adhesion of 2D-3D heterostructures if one can tolerate the altered electronic properties of the material at the defect sites.

Thermal annealing is an established post-process treatment to reduce polymer residues[79,80], remove interfacial water residues[81], and increase the adhesion[9,37]. Thermal annealing is also expected to lead to a more conformal contact of 2D materials with the respective substrates[9,82] and to a larger effective interaction area of the van der Waals forces. We performed button shear tests with hBN on plasma-treated thermal SiO$_2$ samples before and after thermal annealing in N$_2$ atmosphere. The focus was on temperatures from 100 °C to 400 °C, which are typical annealing temperatures compatible with the back end of the line in silicon technology. Anneals up to 1000 °C were additionally executed to investigate possible effects above this regime. The button shear test showed that annealing up to 300 °C – 400 °C significantly increases the adhesion, in line with recent results for graphene[9,38]. Higher temperatures have only a minor additional influence on the adhesion, which may also lie within the standard deviation of the single measurements (Fig. 4a).

We performed scanning thermal microscopy (SThM) to determine the thermal conductivity (or thermal resistance)[83–85] of the multilayer hBN on thermal SiO$_2$ samples before and after annealing as a measure of the conformal contact of the 2D material with its substrate[9,82]. In SThM, the measured thermal signal in Volt corresponds to the thermal interface resistances (TIRs) between the layers for a system of ultrathin layers[86]. This, in turn, correlates with high adhesion as shown for other material systems[87–89]. The thermal signal (V) in the hBN samples decreases significantly from 4.0 V to 3.1 V after a 400 °C anneal (Fig. 4b), but a subsequent anneal at 1000 °C does not further decrease the thermal signal, which remained at 3.2 V. We assume that the TIR between the Si and the thermal SiO$_2$ remained unaffected by annealing at 400 °C and 1000 °C, and attribute the TIR reduction to a change in the hBN and thermal SiO$_2$ interface. This is in good agreement with the adhesion measurement data. Measurements with different probe tips and comparative measurements on a similar sample demonstrate the reproducibility of the results. Additional SThM measurements for hBN on bulk Si samples show a similar behavior (Supplementary Fig. 7).

Neuman et al. showed for graphene that slight deformations occur when the 2D material follows a substrate's roughness, leading to nanometer strain variations[90]. We therefore employed Raman measurements to corroborate our hypothesis that the hBN forms a more conformal contact with the SiO$_2$ substrate upon annealing. Our Raman data of $E_{2g}$ peaks of the hBN on thermal SiO$_2$ before and after thermal annealing show a slight increase of the full-width at half-maximum (FWHM) $\Gamma_{E_{2g}}$ with higher anneal temperatures (histograms in Fig. 4e). This is a signature of increased strain variation, which influences the position of the $E_{2g}$ peak of hBN. The peak broadening is caused by overlapping subpeaks of different strain values in different directions[91], and serves to explain the increased adhesion upon annealing.

## Molybdenum disulfide and tungsten diselenide

We transferred MoS$_2$ and WSe$_2$ with lateral dimensions of approximately one by one centimeter onto thermal SiO$_2$ with O$_2$ plasma cleaning and created buttons on top of and next to the TMDCs. Supplementary Fig. 8 shows Raman spectra of MoS$_2$ and Wse$_2$ films on SiO$_2$. Figure 5 correlates the measured shear strength with the button position on, partly on, or next to the TMDC film. On the WSe$_2$ sample, button column six, row five is on WSe$_2$ and has a $\tau_C = 3.39$ MPa, button column six, row six is partly on WSe$_2$ ($\tau_C = 15.76$ MPa), and button column six, row seven is next to WSe$_2$ and directly on SiO$_2$ ($\tau_C = 34.43$ MPa).

The shear strength on thermal SiO$_2$ with O$_2$ plasma cleaning is $\tau_C = 2.66 \pm 0.09$ MPa and $\tau_C = 3.47 \pm 0.27$ MPa for MoS$_2$ and WSe$_2$, respectively. Adhesion energy values for 2D materials on SiO$_2$ in the literature vary widely. Therefore, only comparisons of two materials with the same measurement method can be used as references. Few publications compare the adhesion of MoS$_2$ on SiO$_2$ with graphene on SiO$_2$. They consistently conclude lower adhesion energy for MoS$_2$ on

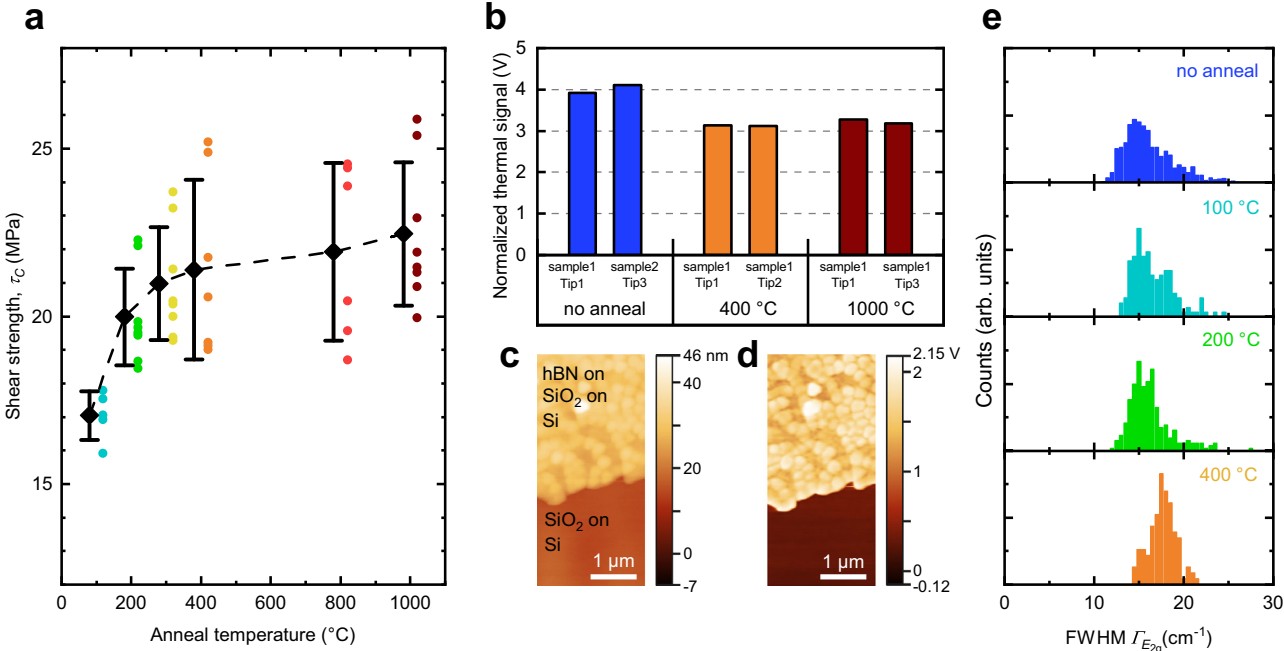

**Fig. 4 | Influence of thermal annealing on a hBN - thermal SiO₂ interface. a** Shear strength $\tau_C$ of monolayer hBN on thermal SiO₂ after different thermal annealing temperatures in N₂ atmosphere. A significant increase of the adhesion until 300 – 400 °C anneal temperature is visible. The average values with the corresponding error bars representing standard deviation are shifted slightly for clarity. The dashed line is a guide to the eye connecting the average values. The number of measured buttons n at the temperatures 100 °C, 200 °C, 300 °C, 400 °C, 800 °C, and 1000 °C is 5, 8, 8, 8, 7, 6, and 8, respectively. **b** Thermal signal of scanning thermal microscopy (SThM) measurements on multilayer hBN on thermal SiO₂ on Si stack before anneal (dark blue), after 400 °C (orange), and after 1000 °C (dark red). The

signal is normalized to the signal of SiO₂ on Si. The thermal resistance is reduced after a 400 °C anneal and remains unchanged after an additional anneal at 1000 °C. Two measurements per temperature are shown to indicate the reproducibility. **c** Exemplary topography map and **d** thermal signal map. These maps were collected on sample 1 after a 400 °C anneal. **e** Full-width at half-maximum (FWHM) $\Gamma_{E_{2g}}$ of hBN Raman signal on monolayer hBN on thermal SiO₂ after different thermal anneal temperatures. Higher anneal temperatures result in a shift towards higher FWHM. The number of Raman measurements on hBN without anneal, after 100 °C anneal, 200 °C, and 400 °C anneal are 1105, 145, 262, and 115, respectively.

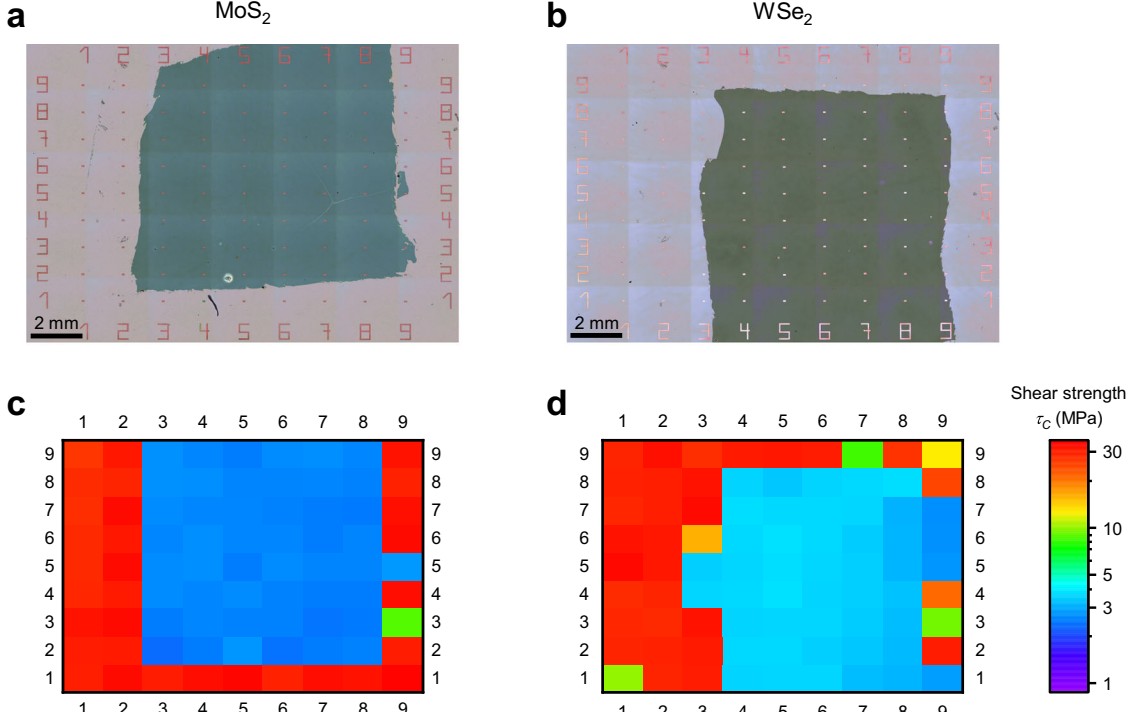

**Fig. 5 | Shear strength $\tau_c$ measurements of MoS₂ and WSe₂.** Optical microscope stitching images of **a** MoS₂ sample and **b** WSe₂ sample. A nine-by-nine array of buttons is measured on both samples. The images are taken after resist development and aluminum etching but before PMMA and 2D material etching. The MoS₂

and WSe₂ films can be identified underneath the PMMA. $\tau_C$ measurements on **c** MoS₂ and **d** WSe₂ sample. $\tau_C$ of each button is displayed in a logarithmic color scale shown on the right side.

$SiO_2$ compared to graphene on $SiO_2$ by blister test[10,14], spontaneously formed blisters[92] intercalation of nanoparticles[18], and AFM techniques[25,93]. The button shear test also assigns a slightly lower shear force to $MoS_2$ compared to graphene ($\tau_C = 2.77 \pm 0.32$ MPa) on thermal $SiO_2$ with $O_2$ plasma cleaning. $\tau_C$ of $WSe_2$ is significantly higher and exhibits a gradient over the film. $\tau_C$ of $WSe_2$ is highest at columns three to five and gradually decreases to columns eight and nine. The reason for this observation remains unclear, although one possible explanation may be non-uniform drying after wet transfer. This data shows that button shear testing can reveal nonuniformities in 2D material samples and may be used to optimize deposition, transfer, and subsequent device fabrication process steps regarding uniformity and adhesion on a wafer scale.

Control of the mechanical properties of the button material is crucial for reproducible results. As discussed previously, the relatively soft PMMA leads to button deformation and pronounced mixed mode near the contacted button edge. Variations in button fabrication can lead to changes in the PMMA properties and, hence, to changes in the measured shear strength[60]. The shear strength of PMMA on thermal $SiO_2$ is $\tau_C = 31.84 \pm 2.61$ MPa on the $MoS_2$ sample and $\tau_C = 30,00 \pm 7,18$ MPa on the $WSe_2$ sample. Previous measurements of PMMA on thermal $SiO_2$ in Fig. 1d with 60 μm buttons ($F_C = 189.75 \pm 1.99$ mN, $\tau_C = 31.63 \pm 0.33$ MPa) and in Fig. 2c with 10 μm s$^{-1}$ ($\tau_C = 34.60$ MPa) performed months before these measurements led to similar $\tau_C$, confirming the reproducibility of button shear testing with PMMA as button material.

Table 2 compares the button shear testing method with other adhesion measurement methods for 2D materials in the literature. It compares favorably in terms of accuracy and measurement effort and provides fraction mode data for 2D materials on standard semiconductor substrates like silicon.

## Discussion

We established button shear testing as a viable and quantitative technique for adhesion measurements of 2D materials parallel to the surface. We established suitable fabrication and measurement parameters for button dimensions and shear speed. The examples of graphene, hBN, $MoS_2$, and $WSe_2$ demonstrate that this approach is suitable for different 2D materials and on various substrates like $Si_3N_4$ and different silicon dioxides. Our results indicate that low surface roughness is beneficial for the strong adhesion of graphene on its substrate. We further used the method to explore means of enhancing adhesion, in particular, $O_2$ plasma treatments of $SiO_2$ surfaces before graphene transfer or post-transfer annealing after a wet transfer of hBN. Our work thus introduces a reliable adhesion measurement technique for 2D materials and provides measurement parameters that can be utilized for quantifying and enhancing the adhesion of 2D materials on rigid substrates. Button shear testing may thus play a crucial role in the development of 2D materials for commercial semiconductor production.

## Methods
### Reference sample
A reference sample without 2D material was fabricated by spin-coating polymethyl methacrylate in chlorobenzene onto an oxidized silicon wafer to create a 5 μm PMMA film. A 20 nm thick aluminum (Al) hard mask was deposited by thermal evaporation. S1805 resist was used to define structures (buttons) with typical lateral dimensions of 60 × 100 μm. The alkaline developer in this process simultaneously etched the Al hard mask. A myplas-III© tool from Plasma Electronic was used to structure the PMMA in an $O_2$ plasma. The buttons were tested both after the removal of the Al film as the last processing step and with the Al film remaining on top of the buttons. We observed no differences in the delamination behavior. The presence of Al provided a higher optical contrast and, hence, faster identification of buttons, so the process option with Al on top of the buttons was chosen for our experiments. A schematic cross-section of the reference sample can be found in Supplementary Fig. 1a. The reference sample was used for button shear test measurements with different button dimensions in Fig. 1d and with different shear speeds in Fig. 2b, c.

### Calibration sample
A calibration sample was fabricated by a lithography process step on blank (100) silicon followed by anisotropic wet etching in potassium hydroxide (KOH) solution. This creates a step in the silicon that can be considered as button that cannot be sheared away. A schematic cross-section of the calibration sample can be found in Supplementary Fig. 1b. The calibration sample was used for button shear test measurements with different cartridges in Fig. 2a.

### Graphene samples for button shear testing
Our graphene samples were based on commercial chemical vapor deposited (CVD) monolayer graphene on copper. This was transferred onto three different substrates by a wet-etching process before button fabrication, (a) 250 nm silicon dioxide deposited with oxygen and tetraethyl orthosilicate precursors (TEOS $SiO_2$), (b) 250 nm silicon nitride deposited from ammonia and dichlorosilane ($Si_3N_4$), and (c) 90 nm silicon dioxide grown by thermal oxidation of silicon (thermal $SiO_2$). A subset of TEOS $SiO_2$ and thermal $SiO_2$ substrates was treated with $O_2$ plasma before graphene transfer. A schematic cross-section of the graphene samples for button shear testing can be found in Supplementary Fig. 1c-g.

**Table 2 | Comparison of adhesion measurement methods of 2D materials**

| Measurement method | Dominant fracture mode | Lateral resolution | Accuracy | Sample preparation effort | Measurement effort |
|---|---|---|---|---|---|
| blister test[10,16] | mixed mode | nm-μm | high | high | high |
| nanoparticles[17,19] | mixed mode | nm-μm | high | moderate | high |
| scratch testing[36,38] | mixed mode | μm | moderate | moderate | low |
| four-point bending[42] | mixed-mode | mm | moderate | moderate | low |
| nanoindentation[24,25] | mode I | nm | high | low | high |
| cantilever beam method[33,34] | mode I | mm | moderate | moderate | low |
| micro force sensing during cleavage[27] | mode II | μm | high | moderate | high |
| AFM friction measurement[23] | mode II | nm | low | low | high |
| substrate stretching on flexible substrates[29–31] | mode II | μm | moderate | low | moderate |
| button shear testing on rigid substrates [this work] | mode II | μm | moderate | moderate | low |

The button shear test method provides viable measurements in fracture mode II with low to moderate effort on rigid substrates like silicon.

### hBN samples for button shear testing

Our hBN samples were based on commercial CVD monolayer hBN, transferred onto $O_2$ plasma-treated thermal $SiO_2$ substrates. Thermal annealing of hBN samples was conducted in nitrogen atmosphere with a hold time of two hours in a Jipelec Jetfirst 300 tool between the transfer process and button fabrication. A schematic cross-section of the hBN samples for button shear testing can be found in Supplementary Fig. 1h-i.

### $MoS_2$ and $WSe_2$ samples for button shear testing

The $MoS_2$ and $WSe_2$ samples were based on MOCVD-grown few-layer $MoS_2$ and $WSe_2$ on sapphire substrates. After spin-coating with PMMA, the TMDC was delaminated from sapphire by $KOH/H_2O$ and transferred onto $O_2$ plasma-treated thermal $SiO_2$ substrates. A schematic cross-section of the $MoS_2$ and $WSe_2$ samples for button shear testing can be found in Supplementary Fig. 1k and Supplementary Fig. 1l, respectively.

### hBN samples for Scanning thermal microscopy (SThM) measurements

SThM samples were prepared by growing 15 nm of thermal oxide on silicon. A lithography step and wet-etching of $SiO_2$ with hydrofluoric acid (HF) created areas with blank Si surface for referencing and normalization of the thermal signal. Multilayer hBN (15 nm) was transferred onto this substrate from copper foil using wet-etching transfer[49]. SThM measurements were performed at the edge of the transferred hBN area to be able to compare the thermal signal with and without hBN. A schematic cross-section and top view of the hBN samples for SThM measurements can be found in Supplementary Fig. 1j.

### Button shear testing

Button shear test measurements were performed with a DAGE4000Plus pull-shear tester from Nordson Corporation. The samples with the processed buttons were placed into the sample holder of the tool. The shear head with a width of 100 to 120 $\mu$m was mounted into a cartridge, positioned 2 $\mu$m above the substrate surface, and a few tens of $\mu$m in front of a button. The stage with the sample moved with a defined speed to create contact between the fixed shear head and the button (Fig. 1c). The lateral displacement of the stage was then recorded while the force that acts on the shear head was measured by the cartridge. The force increased as the shear head made contact with the buttons. The force reached a maximum at a certain point where the shearing of the button started, moving it slightly on the target (see Fig. 1b). After the initiation of the shear process, less force was required to maintain the shear process, and the measurement was stopped. The maximum recorded force corresponds to the force that is required to initiate the shear process. This is expected to occur at the weakest spot of the interface underneath the button and is labeled as critical shear force $F_C$ within this work. Dividing $F_C$ by the button area 100 $\mu$m x 60 $\mu$m leads to the area-shear strength $\tau_C$.

### AFM roughness measurements

AFM roughness measurements were performed on the substrates TEOS $SiO_2$ without $O_2$ plasma, TEOS $SiO_2$ with $O_2$ plasma, $Si_3N_4$, thermal $SiO_2$ without $O_2$ plasma, and thermal $SiO_2$ with $O_2$ plasma and on graphene on TEOS $SiO_2$ without $O_2$ plasma, graphene on $Si_3N_4$, and graphene on thermal $SiO_2$ without $O_2$ plasma with a Bruker Dimension ICON AFM with Nanoscope V Controller. We used ScanAsyst-Air tips with a nominal radius of $r_{tip,nom} = 2$ nm and a spring constant of $k_{nom} = 0.4$ N m$^{-1}$ in ScanAsyst-Mode. Every sample was scanned at two different positions. Before and after scanning the samples, a calibration scan on a PA01 tip characterization sample from MikroMasch Europe was conducted to exclude tip degradation during the scan. The measurements were performed under ambient atmosphere ($T = 21$ °C, $RH = 55$ %).

### XPS measurements

XPS measurements were performed on a Thermo Fisher Scientific ESCALAB Xi$^+$ with an Al K alpha source at 1486.6 eV. TEOS $SiO_2$ samples and thermal $SiO_2$ samples were analyzed by XPS initially, exposed to the $O_2$ plasma afterwards, and finally characterized by XPS again.

### SThM measurements

SThM measurements were performed with a Bruker Dimension ICON AFM with the Nanoscope V Controller. Here, we used a VITA-DM-NANOTA-200 tip and a VITA Module to evaluate the thermal interface resistance (TIR) of hBN to both $SiO_2$ and Si after thermal annealing. Multiple measurements were recorded before and after the annealing process. A plane correction was applied to remove thermal drifts that are common in SThM measurements. The mean value of the thermal signal of the respective area was determined by the software Gwyddion 2.58. This mean thermal signal was normalized such that the thermal signal of Si = 0 V and the thermal signal of $SiO_2$ on Si = 1 V. Constant TIRs between Si and $SiO_2$ are assumed.

### Raman measurements

Raman measurements were performed on hBN before and after anneal with a 532 nm laser, 240 s integration time, and a 100x objective with a Horiba LabRAM Evolution HR system. A linear baseline correction and a fitting with a Lorentz-Gauss profile of the hBN $E_{2g}$ peak and the neon reference lines were performed to evaluate the hBN signal.

## Data availability

Relevant data supporting the key findings of this study are available within the article and the Supplementary Information file. All raw data generated during the current study are available from the corresponding authors upon request.

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

## Acknowledgements

This work has received funding from the European Union's Horizon 2020 research and innovation programme Graphene Flagship Core3 under grant agreement No 881603 (M.C.L., M.H.), from "Bayerisches Staatsministerium für Wirtschaft, Landesentwicklung und Energie StMWi" under the project AlhoiS (DIE-2003-0002//DIEO111/01 (J.S.), DIE-2003-0004//DIEO111/02 (J.W., C.M.)), and from the German Ministry of Education and Research (BMBF) under the projects GIMMIK (03XP0210A (M.H.), 03XP0210F (M.C.L.)), NEUROTEC II (16ME0399 (M.C.L.), 16ME0400 (M.C.L.), 16ME0403 (M.H.)), NeuroSys (03ZU1106AA (M.C.L), 03ZU1106AD (M.H.)), and NobleNEMS (16ES1121K (M.C.L.)).

## Author contributions

M.C.L., S.P., and J.S. designed the study. N.N., A.G., S.L., and J.S. fabricated the samples. N.N., J.Walter and J.S. were involved in the button shear test measurements. A.G., H.K., M.H., and A.V. provided MoS2 and WSe2. J.Weber, S.L., and C.M. conducted the AFM and SThM measurements and S.L. and T.S. the Raman and XPS measurements. M.C.L., J.Walter, S.P., F.S., and J.S. analyzed and discussed the data. The

manuscript was written through contributions of all authors. All authors have given approval for the final version of the manuscript.

## Funding

## Competing interests

The authors declare no competing interests.
