## [Peer Review File · Nature Communications]

Button Shear Testing for Adhesion Measurements of 2D MaterialsREVIEWER COMMENTS

Reviewer #1 (Remarks to the Author):

Authors have reported a shearing technique (called button shear testing in the manuscript) to quantify the critical shear force between 2D materials (graphene and hBN in this work) and treated/untreated substrates (SiO₂ and Si₃N₄). Their results indicate that low surface roughness, oxygen plasma treatment and thermal annealing can be beneficial for the strong adhesion of 2D materials to their underlying substrates. Given that (1) their proposed technique is classified as a relatively well-established mechanical shearing technique and (2) their overall findings would not be highly appealing to the community of 2D materials, I do not recommend this work for publication in Nature Communications.

Below are my main comments on the manuscript:

1) In conventional mechanical shearing techniques, a more solid thin film (e.g., SiO_x, Si₃N₄, Pt/Ir, Pt/Ti) rather than a soft 5- μ m thick PMMA film is used to prevent any deformation and delamination of the top supporting film during the shearing process. Moreover, viscoelastic properties of PMMA can cause the measured data to be travel-speed dependent, and that's why the authors had to perform a set of shear speed-critical shear force measurements to find a more suitable shear speed for their subsequent measurements. To modify their shear testing setup, it is suggested that the authors also read the following work, containing a very well-designed mechanical shearing technique (Nature Communications 6, Article number: 7853 (2015)).

2) CVD-grown hBN and CVD-grown graphene used in this study possess intrinsic defects which can make the interfacial adhesion study even more complicated. In addition to folds, cracks and bilayer patches, Figure S2 also shows some graphene residues left on the sheared region, making discussions on the measured data and comparisons between different case studies even more challenging. Rather, mechanically exfoliated graphene and hBN could be used as a reliable basis for some discussions and comparison purposes in this work. For example, the authors hypothesized that a much higher critical shear force between hBN and SiO₂ (of course, with a much higher standard deviation in Figure 3a) is due to a higher defect density in their CVD-grown hBN, whereas the critical shear force could have been measured between a mechanically exfoliated monolayer graphene/hBN and the underlying substrate to support this hypothesis.

3) The authors also attributed the higher adhesion of graphene on the smoother substrates to the fact that graphene can better follow the substrate morphology of smoother substrates. It is suggested that AFM roughness measurements are performed on the graphene-coated TEOS SiO₂, thermal SiO₂ and Si₃N₄ substrates to compare this new data set with the existing roughness data on the bare substrates.

4) In this work, only the critical shear force has been reported which is a contact surface-dependent quantity. However, the community of 2D materials is more interested in interfacial adhesion energy or interlayer shear strength which are independent of the interfacial contact area.

Reviewer #2 (Remarks to the Author):

Schaiz et. al. developed a method to quantify the adhesion at 2D material/substrate interface using button shear testing method, which has been widely used for adhesion measurement in 3D semiconductor technologies. This is the first time authors use this method for 2D material/substrate interface measurement.

The work appears to be carried out with care and the following comments/concerns should be fully addressed to improve their manuscript.

1. To clarify the adhesion between material and substrate, the shearing must be happened at material/substrate interface. How could authors confirm this. Figure 1b show before and after test but it does not look clear. I suggest authors should show two separate images (before and after test) of the same sample rather than combine them like figure 1b.
2. The adhesion between two materials usually expressed by the term of adhesion energy (Nat. Nanotechnol. 6, 543–546 (2011), Nat. Commun. 11, 5607 (2020), Nature Electronics 6, 146–153 (2023)). But in this work authors express adhesion by critical shear force. If other group use this method with different dimension of button, the critical shear force may be different. Thus, I suggest author convert critical shear force to adhesion energy.
3. To make this work more comprehensive, I suggest to measure adhesion of other TMD materials (MoS₂, WSe₂...) with substrate interface.
4. In the introduction, authors have missed one of methods for measuring interfacial adhesion energy of 2D/2D, 2D/3D and 3D/3D, named 4-point bending. I would recommend author to mention this method in the introduction. The representative publications for this method are Eng. Fract. Mech. 61, 141–162 (1998), Engineering Fracture Mechanics 78, 2390–2398 (2011), Nature Electronics 6, 146–153 (2023).
5. Regarding to sample preparation, was Al film etched after PMMA etching? If so, it should be mentioned somewhere in page #66-70.

Reviewer #3 (Remarks to the Author):

This was an interesting and well written paper showing results of adhesion measurements on 2D materials using a button shear test. The authors establish a new technique for measuring adhesion on 2D materials. This is the first paper that I am aware of that uses button shear testing on 2D materials. They find that surface roughness influences the adhesion with less surface roughness leading to larger adhesion. They also demonstrated that contamination on the surface which can be removed by oxygen plasma treatment influences adhesion. In addition, the authors find that by annealing boron nitride they get a stronger adhesion which they attribute to better conformation of the BN to the substrate. This is corroborated by Raman spectroscopy. The results are timely and likely to be of broad interest to the 2D materials community. The data and analysis in the paper appears sound and sufficient details are provided such that the work can be reproduced. I support publication as is.

Point by point response letter

For the manuscript "Button Shear Testing for Adhesion Measurements of 2D Materials"

Reviewer #1:

Authors have reported a shearing technique (called button shear testing in the manuscript) to quantify the critical shear force between 2D materials (graphene and hBN in this work) and treated/untreated substrates (SiO₂ and Si₃N₄). Their results indicate that low surface roughness, oxygen plasma treatment and thermal annealing can be beneficial for the strong adhesion of 2D materials to their underlying substrates. Given that (1) their proposed technique is classified as a relatively well-established mechanical shearing technique and (2) their overall findings would not be highly appealing to the community of 2D materials, I do not recommend this work for publication in Nature Communications.

Thank you for clearly stating your assessment of our method, with which we partly and politely disagree. (1) The button shear testing method is indeed well-established for conventional materials as a reliable technique used in industry. However, it has never been applied to 2D materials, and had to be specifically adapted to be useful for 2D materials. This is, by definition, novel. We are also convinced that it is (2) **because** the method is established for conventional 3D materials that it will appeal strongly to the 2D materials community. We need reliable, industrially relevant metrology methods for 2D materials to become manufacturable. There is novelty in our manuscript as it describes a simple fabrication route and methodology, including the establishment of measurement parameters, to determine the critical shear force with low to moderate effort. We are convinced that other groups can and will apply our method to assess the adhesion of their 2D materials with the information provided in the manuscript. Therefore, we think that our work provides valuable information, especially for the (nano-) electronics research that now targets the integration of CVD-grown 2D materials into semiconductor manufacturing at the wafer level, as evidenced by numerous industry presentations at the International Electron Device Meeting in San Francisco in December 2023.

- 1) *In conventional mechanical shearing techniques, a more solid thin film (e.g., SiO_x, Si₃N₄, Pt/Ir, Pt/Ti) rather than a soft 5-um thick PMMA film is used to prevent any deformation and delamination of the top supporting film during the shearing process. Moreover, viscoelastic properties of PMMA can cause the measured data to be travel-speed dependent, and that's why the authors had to perform a set of shear speed-critical shear force measurements to find a more suitable shear speed for their subsequent measurements. To modify their shear testing setup, it is suggested that the authors also read the following work, containing a very well-designed mechanical shearing technique (Nature Communications 6, Article number: 7853 (2015)).volume 6, Article number: 7853 (2015)volume 6, Article number).*

We thank the reviewer for this comment. Below we split this comment into several points to answer in detail point by point:

In conventional mechanical shearing techniques, a more solid thin film (e.g., SiO_x, Si₃N₄, Pt/Ir, Pt/Ti) rather than a soft 5-um thick PMMA film is used to prevent any deformation and delamination of the top supporting film during the shearing process.

We chose spin coating as the material deposition technique due to three reasons.

- 1) We had to achieve a minimum button thickness of 4-5 μm to reproducibly contact the button sidewalls at approximately the same (relative) height. This value originates from our engineer's experience with the DAGE4000Plus tester and is not specific to our work. In the

case of thinner buttons, the relative height accuracy of the shear head cannot be guaranteed by our specific tool and the measurement results may be influenced by the shear head height¹. 4-5 μm film thickness can be achieved with spin-coated material. Other semiconductor thin films are often limited to smaller thicknesses due to film stress or low deposition rates.

- 2) Spin coating is applicable to various materials. Other deposition techniques could significantly degrade the 2D material due to elevated temperatures or by through ion damage from plasma processes.
- 3) Spin coating is available in most laboratories while other deposition methods require more sophisticated tools (e.g. PECVD chamber, sputter chamber, CVD furnaces).

We are well aware that there are existing other methodologies for accurate adhesion measurements, as referenced in the submitted manuscript. Our work distinguishes itself by its variability of the substrate material and the 2D material and by the ease of application. Spin coating of the button material is an essential aspect thereof.

In addition, classical applications for button shear tests are mold compounds² but also polyimide is known as a button material³. We want to note that Infineon Technologies AG uses both, mold compound and polyimide as button materials in their tests and routine metrology of conventional 3D materials. Although PMMA and polyimide are different, the material properties are on the same order of magnitude and therefore it is reasonable to use PMMA as button material.

Following your comment, we have also evaluated spin-coated polystyrene as the button material. However, we were not able to structure polystyrene as a cuboid button. The resulting button cross-section of polystyrene is undesired for shear head contacting (see SEM cross sections below). We have added the images to the Supplementary Information file and added a respective sentence to the manuscript:

“Polystyrene as an alternative button material did not lead to a cuboid button cross-section after O₂ plasma-etching and was excluded from the shear experiments (Supplementary Fig. 2).”

We have added the following figure and caption to the Supplementary Information:

Supplementary Fig. 2 Button cross-sections.

a) SEM cross-section of a polystyrene button. The pronounced trapezoidal cross-section is undesired for shear head contacting. b) SEM cross-section of a PMMA button. The cuboid cross-section enables a reliable shear head contacting and button shear testing.

Moreover, viscoelastic properties of PMMA can cause the measured data to be travel-speed dependent, and that's why the authors had to perform a set of shear speed-critical shear force measurements to find a more suitable shear speed for their subsequent measurements.

We agree with the reviewer's observation. In fact, we were aware of the fact that the viscoelastic properties of PMMA lead to shear speed-dependent results and therefore cited, e.g., the paper by Xu et al. in our original submission⁴. This is also why we included our results on the influence of the shear speed in Fig. 2b and Fig. 2c. We then proposed a specific shear speed and kept the shear speed constant for all experiments with 2D materials. This, in fact, can be used as a reference for other groups to establish our method and to use it for adhesion measurements of their own.

For your reference, we repeat the paragraph about PMMA's viscoelastic properties in our original manuscript:

"The influence of the shear speed was evaluated on samples with PMMA on thermal SiO₂ because shear speed effects strongly influence τ_c ⁴⁻⁶. τ_c at low shear speeds below 5 $\mu\text{m s}^{-1}$ was significantly lower than at high shear speeds (Fig. 2b,c). Similar trends were observed in previous studies⁶⁻¹⁰ and can be assigned to viscoelastic properties of PMMA¹¹⁻¹⁴. We chose a shear speed of 10 $\mu\text{m s}^{-1}$ for the experiments with 2D materials to prevent a significant influence of the viscoelastic properties of PMMA."

We believe that we have sufficiently addressed this topic and provided all the information required by other researchers to repeat our experiments.

To modify their shear testing setup, it is suggested that the authors also read the following work, containing a very well-designed mechanical shearing technique (Nature Communications 6, Article number: 7853 (2015)).volume 6, Article number: 7853 (2015)volume 6, Article number).

We appreciate the reviewer's input on the proposed paper Wang, W. et al. Measurement of the cleavage energy of graphite. Nat Commun 6, 7853 (2015). We carefully read the paper, and we consider that publication to be very valuable for the 2D material community. Therefore, we included the method in the introduction:

"A shearing technique for interlayer cleavage energy of graphite was proposed by Wang et al.¹⁵."

We are aware that there are existing methodologies that can provide accurate values for the interfacial adhesion energy that can be inserted into theoretical calculations or simulations. One of them is the mentioned publication by Wang et al. Compared to some of these other methods, our method has a "moderate" and not a "high" accuracy. We made this transparent in Table 2 in

our original manuscript. However, we also note that more sophisticated hardware could decisively enhance the accuracy of our method. Nevertheless, the applicability of our method by other groups strongly depends on the widespread availability of the button shear measurement tool, which is given in many industrial and academic labs. Such groups can easily utilize their commercial button shear testers following the proposed fabrication route. This is important for the next years as 2D material technology matures to higher manufacturing readiness levels. Here, facile wafer-scale metrology methods will be needed for the technology ramp-up in wafer fabs, where the accessibility, repeatability, and ease/speed of use of our method are assets. We have updated Table 2 as follows (marked in red):

Table 2. Comparison of adhesion measurement methods of 2D materials. The button shear test method provides viable measurements in fracture mode II with low to moderate effort on rigid substrates like silicon.

measurement method	dominant fracture mode	lateral resolution	accuracy	sample preparation effort	measurement effort
blister test ^{16,17}	mixed mode	nm- μm	high	high	high
nanoparticles ^{18,19}	mixed mode	nm- μm	high	moderate	high
scratch testing ^{20,21}	mixed mode	μm	moderate	moderate	low
Four-point bending²²	mixed-mode	mm	moderate	moderate	low
nanoindentation ^{23,24}	mode I	nm	high	low	high
cantilever beam method ^{25,26}	mode I	mm	moderate	moderate	low
Micro force sensing during cleavage¹⁵	mode II	μm	high	moderate	high
AFM friction measurement ²⁷	mode II	nm	low	low	high
substrate stretching on flexible substrates ^{5,28,29}	mode II	μm	moderate	low	moderate
button shear testing on rigid substrates ^{this work}	mode II	μm	moderate	moderate	low

- 2) *CVD-grown hBN and CVD-grown graphene used in this study possess intrinsic defects which can make the interfacial adhesion study even more complicated. In addition to folds, cracks and bilayer patches, Figure S2 also shows some graphene residues left on the sheared region, making discussions on the measured data and comparisons between different case studies even more challenging. Rather, mechanically exfoliated graphene and hBN could be used as a reliable basis for some discussions and comparison purposes in this work. For example, the authors hypothesized that a much higher critical shear force between hBN and SiO₂ (of course, with a much higher standard deviation in Figure 3a) is due to a higher defect density in their CVD-grown*

hBN, whereas the critical shear force could have been measured between a mechanically exfoliated monolayer graphene/hBN and the underlying substrate to support this hypothesis.

Thank you for pointing this out. Using mechanically exfoliated 2D material is, in principle, a very reasonable proposal. However, it will be practically difficult or impossible to exfoliate sufficient monolayer graphene and hBN flakes with lateral dimensions of 60 x 100 μm to achieve statistics in button shear testing. In addition, the focus of our work is not on providing the highest measurement accuracy using ideal (e.g. exfoliated) material systems, but on the practicable assessment of state-of-the-art wafer-scale 2D materials in nano- and optoelectronics research, which is CVD-grown material rather than exfoliated material. Therefore, we decided to stick to CVD-grown 2D material in this manuscript.

However, after receiving this feedback, we have added more information on our graphene on thermal SiO_2 and hBN on thermal SiO_2 in Supplementary Fig. 6. We have applied laser microscopy to illustrate the density and lateral size of bi- and multilayer areas. We have also conducted AFM measurements to reveal even small folds on graphene and to visualize multilayer areas on hBN. Finally, we have performed scanning Raman measurements on graphene and compared it with the already existing Raman data of hBN (from Fig. 4e) to get an impression of the defect density on the nanometer scale. Based on these measurements, we can conclude a higher defect density in our hBN compared to our graphene, which supports the hypothesis of a higher adhesion with higher defect density, as also suggested in the literature^{29,30}. We have added a respective sentence to the manuscript:

“As an indication of the defect density of graphene and hBN used in this work, we provide laser scanning microscope images, AFM measurements, and Raman measurements in Supplementary Fig 6.”

We have added the following figure and caption to the Supplementary Information:

Supplementary Fig. 6 Investigation of graphene and hBN on thermal SiO₂.

Laser scanning microscope images of a) graphene and b) hBN on thermal SiO₂. The graphene in this work consists of predominantly monolayer graphene with a few bi- and multi-layer areas with a lateral size of approx. 10 μm and some folds. The hBN in this work consists of monolayer hBN with a high density of multilayer areas with a lateral size of approx. 1 μm. AFM measurements of c) graphene and d) hBN. Some additional minor folds on the graphene monolayer area become visible. The hBN multilayer areas are detected with a height of up to 10 nm thickness. Exemplary Raman measurements of e) graphene and f) hBN. No D peak was detected on graphene and an I_{2D}/I_G ratio of 1.08 on the monolayer area and 0,53 on the bilayer area is found. Measurements on three arbitrary points on hBN reveal a small E_{2g} peak with varying intensity. Statistical distribution of the FWHM of g) 2D peaks of graphene and h) E_{2g} peaks of hBN.

3) The authors also attributed the higher adhesion of graphene on the smoother substrates to the fact that graphene can better follow the substrate morphology of smoother substrates. It is suggested that AFM roughness measurements are performed on the graphene-coated TEOS SiO₂, thermal SiO₂ and Si₃N₄ substrates to compare this new data set with the existing roughness data on the bare substrates.

We thank the reviewer for this suggestion. We have measured the roughness of graphene on TEOS SiO₂, Si₃N₄, and thermal SiO₂ and include it here as Response Letter Fig. 1. We have also added the roughness of graphene on the three substrates to our original Table 1.

Response Letter Fig. 1 AFM measurements of graphene on a) TEOS SiO₂ with $s_a = 0.64$ nm, b) Si₃N₄ with $s_a = 0.42$ nm, and c) thermal SiO₂ with $s_a = 0.22$ nm. The white rectangles indicate the area for roughness determination.

Table 1. Comparison of contact angle measurements, roughness measurements, and shear strengths τ_c of graphene on TEOS SiO₂, Si₃N₄, and thermal SiO₂ and of hBN on thermal SiO₂.

	TEOS SiO ₂		Si ₃ N ₄	Thermal SiO ₂	
O ₂ plasma	No	Yes	No	No	Yes
CA_{water} (°)	7.3±1.5	3.0±1.3	19.3±0.3	42.7±0.3	2.7±1.6
$CA_{diiodomethane}$ (°)	40.6±0.4	35.3±0.6	40.7±1.8	45.2±0.5	34.6±0.3
Surface Energy (mN m ⁻¹)	76.0±0.6	77.4±1.9	73.0±0.6	60.6±0.2	77.5±1.8
Roughness s_a of the substrate (nm)	0.76	0.69	0.30	0.22	0.21
Roughness s_a of graphene on the substrate (nm)	0.64		0.42	0.22	
Shear strength τ_c of graphene (MPa)	1.55±0.31	2.21±0.33	2.88±0.31	2.68±0.11	2.77±0.32
Shear strength τ_c of hBN (MPa)					4.43±0.95

On thermal SiO₂ the roughness is identical ($s_a = 0.22$ nm with and without graphene), while on Si₃N₄ the roughness is higher with graphene ($s_a = 0.42$ nm with graphene and $s_a = 0.30$ nm without graphene). A definite explanation for the higher roughness on Si₃N₄ is not possible with the available data. One potential reason may be PMMA residues from the wet-etching

transfer process^{31,32}. However, it remains unclear why the roughness is not increased on thermal SiO₂. Typically, these polymer residues are removed by thermal annealing^{33,34} but this would change the morphology of the graphene³⁵ and therefore cannot be applied in our case.

On TEOS SiO₂ the roughness is lower with graphene ($s_a = 0.64$ nm with graphene and $s_a = 0.76$ nm without graphene). This fits predictions from literature^{36–38} and our statement in the original manuscript that graphene can follow the substrate morphology less on the rough TEOS substrate.

We describe these results in the revised manuscript as follows:

Fig. 3a compares τ_c of graphene on as deposited TEOS SiO₂ (1.55 ± 0.31 MPa), Si₃N₄ (2.88 ± 0.31 MPa), and thermal SiO₂ (2.68 ± 0.11 MPa). We correlated these results with the surface roughness s_a of the substrates and the surface roughness of graphene on the substrates as the roughness is known to influence the adhesion of a 2D material^{39,40}. AFM roughness data show that bare TEOS SiO₂ has a roughness of $s_a = 0.76$ nm and graphene on TEOS SiO₂ has a lower roughness of $s_a = 0.64$ nm. On the smoother substrate thermal SiO₂ the roughness remains at $s_a = 0.22$ nm without and with graphene. On Si₃N₄ the roughness increases from $s_a = 0.30$ nm without graphene to $s_a = 0.42$ nm with graphene. This increase in roughness may be attributed to PMMA residues from the wet etching transfer process^{31,32} and limits the validity of roughness measurements on transferred CVD-grown graphene. However, the lower roughness of graphene on TEOS SiO₂ compared to bare TEOS SiO₂ is an indication that the graphene does not fully follow the TEOS SiO₂ morphology. This behavior on rough substrates is predicted in theoretical studies^{36,37} and leads to a smaller effective interface area and finally to a smaller adhesion³⁸.

- 4) *In this work, only the critical shear force has been reported which is a contact surface-dependent quantity. However, the community of 2D materials is more interested in interfacial adhesion energy or interlayer shear strength which are independent of the interfacial contact area.*

Thank you very much for this feedback. After the feedback of reviewer #1 comment 4 and reviewer #2 comment 2 (see below), we decided to change the critical shear force F_c (in mN) to the area normalized shear strength τ_c (in MPa) from Figure 2c onwards throughout the manuscript. These changes were performed in the manuscript as follows:

Abstract:

We show with our quantitative data that low substrate roughness and oxygen plasma treatments on the substrates before 2D material transfer result in higher shear strengths.

Introduction:

Here, we introduce button shear testing as a quantitative method for determining the shear strengths of 2D materials.

Results, Button shear testing:

[...] Therefore, no quantitative evaluation of the displacement was performed, but F_c was extracted only from the force data. Dividing F_c by the button area $100 \mu\text{m} \times 60 \mu\text{m}$ leads to the area independent shear strength τ_c .

The influence of the shear speed was evaluated on samples with PMMA on thermal SiO₂, because shear speed effects strongly influence τ_c ^{4–6}. τ_c at low shear speeds below $5 \mu\text{m s}^{-1}$ was significantly lower than at high shear speeds (Fig. 2b,c). [...]

Results, Graphene:

The **shear strength** of graphene on the three substrates TEOS SiO₂, Si₃N₄, and thermal SiO₂ was measured with the optimized parameters of a small cartridge and 10 μm s⁻¹ shear speed. Fig. 3a compares τ_c of graphene on as deposited TEOS SiO₂ (**1.55±0.31 MPa**), Si₃N₄ (**2.88±0.31 MPa**), and thermal SiO₂ (**2.68±0.11 MPa**).

[...]

The O₂ plasma treatment increased τ_c of graphene on TEOS SiO₂ to **2.21±0.33 MPa**, but we did not observe a significant change on thermal SiO₂ (**2.77±0.32 MPa**, Fig. 3a).

[...]

Since the reduction of CA_{water} is more pronounced on thermal SiO₂, which showed no significant change in τ_c , we rule this effect out as the main contributor.

[...]

Results, Hexagonal boron nitride:

[...] The τ_c of **4.43±0.95 MPa** is significantly higher on this substrate compared to graphene, which was **2.77±0.32 MPa**, [...]

Methods, Button shear testing:

Dividing F_c by the button area 100 μm x 60 μm leads to the area-shear strength τ_c .

Figure 2:

Fig. 2 Characterization of cartridge and influence of shear speed. **a** Force vs. displacement curves of measurements with two different cartridges contacting a stable obstacle. The small cartridge delivers a lower noise in the force signal at the cost of no reliable displacement data. **b** Force vs. displacement curves and **c** extracted F_c and τ_c on samples with PMMA on SiO₂ at different shear velocities from 1 μm s⁻¹ (blue) to 50 μm s⁻¹ (red). The velocity 10 μm s⁻¹ (orange) is highlighted and was used in further experiments with 2D materials.

Figure 3:

Fig. 3 Characterization of TEOS SiO_2 , Si_3N_4 , and thermal SiO_2 substrates and shear strength determination of graphene and hBN on the substrates. a τ_c of graphene on TEOS SiO_2 (green, number of buttons $n = 11$), Si_3N_4 (red, $n = 9$), and thermal SiO_2 (blue, $n = 5$). The right blue box plot represents the τ_c of hBN on thermal SiO_2 ($n = 8$). [...]

Figure 4:

Fig. 4 Influence of thermal annealing on a hBN - thermal SiO₂ interface. a τ_c of monolayer hBN on thermal SiO₂ after different thermal annealing temperatures in N₂ atmosphere. A significant increase of the adhesion until 300 – 400 °C anneal temperature is visible. [...]

Table 1:

Table 1. Comparison of contact angle measurements, roughness measurements, and shear strengths τ_c of graphene on TEOS SiO₂, Si₃N₄, and thermal SiO₂ and of hBN on thermal SiO₂.

	TEOS SiO ₂		Si ₃ N ₄	Thermal SiO ₂	
O ₂ plasma	No	Yes	No	No	Yes
CA_{water} (°)	7.3±1.5	3.0±1.3	19.3±0.3	42.7±0.3	2.7±1.6
$CA_{diiodomethane}$ (°)	40.6±0.4	35.3±0.6	40.7±1.8	45.2±0.5	34.6±0.3
Surface Energy (mN m ⁻¹)	76.0±0.6	77.4±1.9	73.0±0.6	60.6±0.2	77.5±1.8
Roughness s_a of the substrate (nm)	0.76	0.69	0.30	0.22	0.21
Roughness s_a of graphene on the substrate (nm)	0.64		0.42	0.22	
Shear strength τ_c of graphene (MPa)	1.55±0.31	2.21±0.33	2.88±0.31	2.68±0.11	2.77±0.32
Shear strength τ_c of hBN (MPa)					4.43±0.95

Reviewer #2:

- 1) To clarify the adhesion between material and substrate, the shearing must be happened at material/substrate interface. How could authors confirm this. Figure 1b show before and after test but it does not look clear. I suggest authors should show two separate images (before and after test) of the same sample rather than combine them like figure 1b.

We thank reviewer #2 for this comment. We agree that the verification of delamination at the substrate-2D material interface was not sufficient in the original version. To keep the manuscript concise, we performed only a minor change in Fig. 1b and added Supplementary Information Figures instead. Supplementary Fig. 2 showed the same buttons before and after button shear testing already in the originally submitted version (in the revised Supplementary Information it is Supplementary Fig. 3).

The new Supplementary Fig. 4 shows optical microscope and laser scanning microscope images of exemplary sheared interfaces for each sample. They confirm that delamination predominantly occurred at the substrate-2D material interface on all samples.

The new Supplementary Fig. 5 shows a Raman map of an exemplary sheared interface of graphene on thermal SiO₂ without O₂ plasma treatment. This confirms that the darker areas in Supplementary Fig. 4 are graphene residues and no graphene is detected in a few μm distance from the contacted edge.

Fig. 1b has changed to:

Fig. 1 Illustration of button geometry and shear test mechanism. [...]

We have added the following statement in the manuscript in section Results, Graphene in addition to the two new Figures in the Supplementary Information, which are also shown below for your reference.

“In all cases, delamination at the substrate-graphene interface occurred. This was confirmed by optical microscopy (Supplementary Fig. 4) and Raman measurements (Supplementary Fig. 5).”

Supplementary Fig. 4 Microscope images of sheared interfaces.

Optical microscope images of sheared interfaces after button shear testing. a) Graphene on thermal SiO₂ without O₂ plasma treatment and b) graphene on thermal SiO₂ with O₂ plasma treatment. Graphene is sheared from the thermal SiO₂ substrate at the major part of the button area. Minor areas with graphene remain on the surface and are visible as dark blue areas, especially on the contacted button edge (top edge in the images). Laser scanning microscope images of c) graphene on TEOS SiO₂ without O₂ plasma treatment, d) graphene on TEOS SiO₂ with O₂ plasma treatment, e) graphene on Si₃N₄, f) hBN on thermal SiO₂ with O₂ plasma treatment, g) MoS₂ on thermal SiO₂ with O₂ plasma treatment, and h) WSe₂ on thermal SiO₂ with O₂ plasma treatment. On all samples, only minor 2D material residues are detected. Delamination occurred predominantly at the substrate-2D material interface.

Supplementary Fig. 5 Raman measurement of sheared interface graphene-thermal SiO₂.

a) Overlap of optical microscope image and G peak intensity heat map. The original button edge is indicated as a white dashed line and the shear direction of the shear head as a white arrow. b) 2D peak intensity of the same area. Graphene residues remain at the contacted edge and nearly no graphene is detected at a few μm distance from the contacted edge.

- 2) *The adhesion between two materials usually expressed by the term of adhesion energy (Nat. Nanotechnol. 6, 543–546 (2011), Nat. Commun. 11, 5607 (2020), Nature Electronics 6, 146–153 (2023)). But in this work authors express adhesion by critical shear force. If other group use this method with different dimension of button, the critical shear force may be different. Thus, I suggest author convert critical shear force to adhesion energy.*

Thank you very much for this feedback. After the feedback of reviewer #1 comment 4 and reviewer #2 comment 2, we decided to change the critical shear force F_c (in mN) to the area normalized shear strength τ_c (in MPa) from Fig. 2c onwards throughout the manuscript.

However, a conversion to adhesion energy (in J/m²) is not possible. To determine the adhesion energy by shear tests, the force needs to be integrated over the displacement. A reliable distance measurement is very challenging even for buttons in the millimeter range⁴¹. As described in the manuscript in Fig. 2a and 2b, we were not able to perform a reliable distance measurement. This may be possible with future research.

The following changes were made in the manuscript in the respective sections (the list is repeated here from above for your reference):

Abstract:

We show with our quantitative data that low substrate roughness and oxygen plasma treatments on the substrates before 2D material transfer result in higher **shear strengths**.

Introduction:

Here, we introduce button shear testing as a quantitative method for determining the **shear strengths** of 2D materials.

Results, Button shear testing:

[...] Therefore, no quantitative evaluation of the displacement was performed, but F_c was extracted only from the force data. **Dividing F_c by the button area 100 μm x 60 μm leads to the area independent shear strength τ_c .**

The influence of the shear speed was evaluated on samples with PMMA on thermal SiO₂, because shear speed effects strongly influence τ_c ^{4–6}. τ_c at low shear speeds below 5 $\mu\text{m s}^{-1}$ was significantly lower than at high shear speeds (Fig. 2b,c). [...]

Results, Graphene:

The **shear strength** of graphene on the three substrates TEOS SiO₂, Si₃N₄, and thermal SiO₂ was measured with the optimized parameters of a small cartridge and 10 $\mu\text{m s}^{-1}$ shear speed. Fig. 3a compares τ_c of graphene on as deposited TEOS SiO₂ (**1.55±0.31 MPa**), Si₃N₄ (**2.88±0.31 MPa**), and thermal SiO₂ (**2.68±0.11 MPa**).

[...]

The O₂ plasma treatment increased τ_c of graphene on TEOS SiO₂ to **2.21±0.33 MPa**, but we did not observe a significant change on thermal SiO₂ (**2.77±0.32 MPa**, Fig. 3a).

[...]

Since the reduction of CA_{water} is more pronounced on thermal SiO₂, which showed no significant change in τ_c , we rule this effect out as the main contributor.

[...]

Results, Hexagonal boron nitride:

[...] The τ_c of **4.43±0.95 MPa** is significantly higher on this substrate compared to graphene, which was **2.77±0.32 MPa**, [...]

Methods, Button shear testing

Dividing F_C by the button area $100\ \mu\text{m} \times 60\ \mu\text{m}$ leads to the area-shear strength τ_c .

Figure 2:

Fig. 2 Characterization of cartridge and influence of shear speed. **a** Force vs. displacement curves of measurements with two different cartridges contacting a stable obstacle. The small cartridge delivers a lower noise in the force signal at the cost of no reliable displacement data. **b** Force vs. displacement curves and **c** extracted F_C and τ_c on samples with PMMA on SiO_2 at different shear velocities from $1\ \mu\text{m s}^{-1}$ (blue) to $50\ \mu\text{m s}^{-1}$ (red). The velocity $10\ \mu\text{m s}^{-1}$ (orange) is highlighted and was used in further experiments with 2D materials.

Figure 3:

Fig. 3 Characterization of TEOS SiO₂, Si₃N₄, and thermal SiO₂ substrates and shear strength determination of graphene and hBN on the substrates. a τ_c of graphene on TEOS SiO₂ (green, number of buttons $n = 11$), Si₃N₄ (red, $n = 9$), and thermal SiO₂ (blue, $n = 5$). The right blue box plot represents the τ_c of hBN on thermal SiO₂ ($n = 8$). [...]

Figure 4:

Fig. 4 Influence of thermal annealing on a hBN - thermal SiO₂ interface. a τ_c of monolayer hBN on thermal SiO₂ after different thermal annealing temperatures in N₂ atmosphere. A significant increase of the adhesion until 300 – 400 $^{\circ}\text{C}$ anneal temperature is visible. [...]

Table 1:

Table 1. Comparison of contact angle measurements, roughness measurements, and shear strengths τ_c of graphene on TEOS SiO₂, Si₃N₄, and thermal SiO₂ and of hBN on thermal SiO₂.

	TEOS SiO ₂		Si ₃ N ₄	Thermal SiO ₂	
O ₂ plasma	No	Yes	No	No	Yes
CA_{water} ($^{\circ}$)	7.3 \pm 1.5	3.0 \pm 1.3	19.3 \pm 0.3	42.7 \pm 0.3	2.7 \pm 1.6
$CA_{\text{diiodomethane}}$ ($^{\circ}$)	40.6 \pm 0.4	35.3 \pm 0.6	40.7 \pm 1.8	45.2 \pm 0.5	34.6 \pm 0.3
Surface Energy (mN m ⁻¹)	76.0 \pm 0.6	77.4 \pm 1.9	73.0 \pm 0.6	60.6 \pm 0.2	77.5 \pm 1.8
Roughness s_a of the substrate (nm)	0.76	0.69	0.30	0.22	0.21
Roughness s_a of graphene on the substrate (nm)	0.64		0.42	0.22	
Shear strength τ_c of graphene (MPa)	1.55 \pm 0.31	2.21 \pm 0.33	2.88 \pm 0.31	2.68 \pm 0.11	2.77 \pm 0.32
Shear strength τ_c of hBN (MPa)					4.43 \pm 0.95

- 3) To make this work more comprehensive, I suggest to measure adhesion of other TMD materials (MoS₂, WSe₂...) with substrate interface.

We thank reviewer #2 for this suggestion. We have performed **additional experiments with the MOCVD-grown transition metal dichalcogenides (TMDCs) MoS₂ and WSe₂**. We have used a new mask layout on these samples with more buttons of the defined size (100 μm x 60 μm), based on the experience gained in our original experiments. With this mask, it is possible to correlate the shear strength τ_c of each button with its lateral location on the TMDCs as shown in the new Figure 5. We also used this mask to measure the adhesion of the buttons on the original substrates, i.e. next to the TMDCs (see details below). The shear strength of MoS₂ on thermal SiO₂ with O₂ plasma cleaning is $\tau_c = 2,66 \pm 0,09$ MPa and that of WSe₂ on thermal SiO₂ with O₂ plasma cleaning is $\tau_c = 3,47 \pm 0,27$ MPa. A very limited number of references is available but a comparison of MoS₂ on SiO₂ with graphene is available^{16,24,42–45}. According to the results obtained by other groups using other methods, the adhesion of MoS₂ on SiO₂ is lower than that of graphene on SiO₂. This is confirmed by our results, but the difference we measure is rather small and within the uncertainty ($\tau_c = 2,77 \pm 0,32$ MPa for graphene). We are not aware of any literature that compares the adhesion energy or the shear strength of WSe₂ on SiO₂ with MoS₂ or graphene. We can, therefore, only speculate about the reason for the higher shear strength of WSe₂. In Fig. 5d, a gradual decrease of the adhesion energy is visible from left (columns 3, 4) to right (columns 8, 9). Most parameters (SiO₂ roughness, SiO₂ surface contamination, quality of MOCVD-grown material) do not typically change gradually in a range of a few millimeters. Therefore, we speculate that non-uniform drying after the wet transfer could be a possible reason, but also declare that the reason remains unclear.

The reproducibility of the measurement method after several months can be illustrated with these additional measurements. The shear strength of PMMA on thermal SiO₂ measured as a reference is $\tau_c = 31,84 \pm 2,61$ MPa on the MoS₂ sample and $\tau_c = 30,00 \pm 7,18$ MPa on the WSe₂ sample. This was possible because the 2D materials did not cover the entire chips and hence, there were several PMMA buttons directly on the SiO₂ surface. This shear strength is consistent with the previous measurements of PMMA on thermal SiO₂ in Fig. 1d with 60 μm buttons ($F_c = 189,75 \pm 1,99$ mN, $\tau_c = 31,63 \pm 0,33$ MPa) and in Fig. 2c with 10 μm s⁻¹ ($\tau_c = 34,60$ MPa).

The high standard deviation on the WSe₂ sample mainly originates from 4 outliers with low adhesion of 8 to 13 MPa and may be attributed to contamination during the transfer process.

MoS₂ and WSe₂ was provided by the Compound Semiconductor Technology group of RWTH Aachen University and AIXTRON SE, Aachen, and partially processed by a researcher at the Chair of Electronic Devices at RWTH Aachen University. Adding these experiments led to the inclusion of additional coauthors and their affiliations, as well as an update to the author contribution statements and the acknowledgments:

Josef Schätz^{1,2}, Navin Nayi^{1,#}, Jonas Weber^{3,4}, Christoph Metzke^{3,5}, Sebastian Lukas², Jürgen Walter¹, Tim Schaffus¹, Fabian Streb¹, Annika Grundmann⁶, Holger Kalisch⁶, Michael Heuken^{6,7}, Andrei Vescan⁶, Stephan Pindl¹, and Max C. Lemme^{2,8}*

¹Infinion Technologies AG, Wernerwerkstraße 2, 93049 Regensburg, Germany

²Chair of Electronic Devices, RWTH Aachen University, Otto-Blumenthal-Str. 25, 52074 Aachen, Germany

³Department of Electrical Engineering and Media Technology, Deggendorf Institute of Technology, Dieter-Görlitz-Platz 1, 94469 Deggendorf, Germany

⁴Department of Applied Physics, University of Barcelona, Martí i Franquès 1, 08028 Barcelona, Spain

⁵Department of Electrical Engineering, Helmut Schmidt University/University of the Federal Armed Forces Hamburg, Holstenhofweg 85, 22043 Hamburg, Germany

⁶Compound Semiconductor Technology, RWTH Aachen University, Sommerfeldstr. 18, 52074 Aachen, Germany

⁷AIXTRON SE, Dornkaulstr. 2, 52134 Herzogenrath, Germany

⁸AMO GmbH, Advanced Microelectronic Center Aachen, Otto-Blumenthal-Str. 25, 52074 Aachen, Germany

Author Contributions

M.C.L., P.S., and J.S. designed the study. N.N., A.G., S.L., and J.S. fabricated the samples. N.N., J.Walter and J.S. were involved in the button shear test measurements. A.G., H.K., M.H., and A.V. provided MoS₂ and WSe₂. J.Weber, S.L., and C.M. conducted the AFM and SThM measurements and S.L. and T.S. the Raman and XPS measurements. M.C.L., J.Walter, P.S., F.S., and J.S. analyzed and discussed the data. The manuscript was written through contributions of all authors. All authors have given approval for the final version of the manuscript.

Acknowledgements

This work has received funding from the European Union's Horizon 2020 research and innovation program under grant agreement 881603 (Graphene Flagship Core3), from "Bayerisches Staatsministerium für Wirtschaft, Landesentwicklung und Energie StMWi" (DIE-2003-0004//DIE0111/02) and from the German Ministry of Education and Research (BMBF) under the projects GIMMIK (03XP0210A, 03XP0210F), NEUROTEC 2 (16ME0399, 16ME0400, 16ME0403), and NeuroSys (03ZU1106AA, 03ZU1106AD).

The new results on MoS₂ and WSe₂ have been added to the revised manuscript as follows:

Abstract:

To this end, button shear testing is proposed and demonstrated as a method for evaluating the adhesion of 2D materials with the examples of graphene, hexagonal boron nitride (hBN), molybdenum disulfide, and tungsten diselenide on silicon dioxide and silicon nitride substrates.

Results, Button shear testing:

CVD hexagonal boron nitride (hBN) was also transferred from a copper growth substrate onto O₂ plasma-treated thermal SiO₂ substrates by a wet-etching process⁴⁶, and subsets with hBN were annealed up to 1000 °C between the transfer process and button fabrication. Metal-organic chemical-vapor-deposited (MOCVD) molybdenum disulfide (MoS₂)^{47,48} and MOCVD tungsten diselenide (WSe₂)⁴⁹ was transferred from the sapphire growth substrate onto O₂ plasma-treated thermal SiO₂ substrates.

Result, New subheading:

Molybdenum disulfide and tungsten diselenide

We transferred MoS₂ and WSe₂ with lateral dimensions of approximately one by one centimeter onto thermal SiO₂ with O₂ plasma cleaning and created buttons on top of and next to the TMDCs. Supplementary Fig. 8 shows Raman spectra of MoS₂ and WSe₂ films on SiO₂. Fig. 5 correlates the measured shear strength with the button position on, partly on, or next to the TMDC film. On the WSe₂ sample, button column six, row five is on WSe₂ and has a $\tau_c = 3.39$ MPa, button column six, row six is partly on WSe₂ ($\tau_c = 15.76$ MPa), and button column six, row seven is next to WSe₂ and directly on SiO₂ ($\tau_c = 34.43$ MPa).

The shear strength on thermal SiO₂ with O₂ plasma cleaning is $\tau_c = 2.66 \pm 0.09$ MPa and $\tau_c = 3.47 \pm 0.27$ MPa for MoS₂ and WSe₂, respectively. Adhesion energy values for 2D materials on SiO₂ in the literature vary widely. Therefore, only comparisons of two materials with the same measurement method can be used as references. Few publications compare the adhesion of MoS₂ on SiO₂ with graphene on SiO₂. They consistently conclude lower adhesion energy for MoS₂ on SiO₂ compared to graphene on SiO₂ by blister test^{16,45}, spontaneously formed blisters⁴² intercalation of nanoparticles⁴³, and AFM techniques^{24,44}. The button shear test also assigns a slightly lower shear force to MoS₂ compared to graphene ($\tau_c = 2.77 \pm 0.32$ MPa) on thermal SiO₂ with O₂ plasma cleaning. τ_c of WSe₂ is significantly higher and exhibits a gradient over the film. τ_c of WSe₂ is highest at columns three to five and gradually decreases to columns eight and nine. The reason for this observation remains unclear, although one possible explanation may be non-uniform drying after wet transfer. This data shows that button shear testing can reveal nonuniformities in 2D material samples and may be used to optimize deposition, transfer, and subsequent device fabrication process steps regarding uniformity and adhesion on a wafer scale.

The shear strength of PMMA on thermal SiO₂ is $\tau_c = 31.84 \pm 2.61$ MPa on the MoS₂ sample and $\tau_c = 30.00 \pm 7.18$ MPa on the WSe₂ sample. Previous measurements of PMMA on thermal SiO₂ in Fig. 1d with 60 μm buttons ($F_c = 189.75 \pm 1.99$ mN, $\tau_c = 31.63 \pm 0.33$ MPa) and in Fig. 2c with 10 $\mu\text{m s}^{-1}$ ($\tau_c = 34.60$ MPa) performed months before these measurements led to similar τ_c , confirming the reproducibility of button shear testing.

Fig. 5 Shear strength measurements of MoS₂ and WSe₂. Optical microscope stitching images of **a** MoS₂ sample and **b** WSe₂ sample. A nine-by-nine array of buttons is measured on both samples. The images are taken after resist development and aluminum etching but before PMMA and 2D material etching. The MoS₂ and WSe₂ films can be identified underneath the PMMA. τ_c measurements on **c** MoS₂ and **d** WSe₂ sample. The shear strength of each button is displayed in a logarithmic color scale shown on the right side.

Discussion:

The examples of graphene, hBN, MoS₂, and WSe₂ demonstrate that this approach is suitable for different 2D materials and on various substrates like Si₃N₄ and different silicon dioxides.

Methods:

MoS₂ and WSe₂ samples for button shear testing

The MoS₂ and WSe₂ samples were based on MOCVD-grown few-layer MoS₂ and WSe₂ on sapphire substrates. After spin-coating with PMMA, the TMDC was delaminated from sapphire by KOH/H₂O and transferred onto O₂ plasma-treated thermal SiO₂ substrates. A schematic cross-section of the MoS₂ and WSe₂ samples for button shear testing can be found in Supplementary Fig. 1k and Supplementary Fig. 1l, respectively.

We have further added the new processing steps to the comprehensive summary of the samples and fabrication processes in the Supplementary Information:

Supplementary Fig. 1 Schematic cross sections of all samples and their specific fabrication pretreatments.

Schematic cross sections of a) reference samples with PMMA button on 90 nm thermal SiO_2 to define button dimensions and shear speed and b) calibration samples with a step in silicon to define the button shear tester cartridge. Samples with graphene on c) TEOS SiO_2 , d) Si_3N_4 , and e) thermal SiO_2 . Samples with graphene and O_2 plasma as pretreatment on f) TEOS SiO_2 and g) thermal SiO_2 . Sample with hBN on thermal SiO_2 with O_2 plasma as pretreatment and h) without and i) with thermal anneal after hBN transfer. j) Samples for SThM measurement with hBN on 15 nm thermal SiO_2 . Sample with k) MoS_2 and l) WSe_2 on thermal SiO_2 with O_2 plasma as pretreatment.

Supplementary Fig. 8 Raman spectra of a) MoS₂ and b) WSe₂.

- 4) In the introduction, authors have missed one of methods for measuring interfacial adhesion energy of 2D/2D, 2D/3D and 3D/3D, named 4-point bending. I would recommend author to mention this method in the introduction. The representative publications for this method are *Eng. Fract. Mech.* 61, 141–162 (1998), *Engineering Fracture Mechanics* 78, 2390–2398 (2011), *Nature Electronics* 6, 146–153 (2023).

We appreciate the reviewer’s input, as we had not been aware of 4-point bending to measure 2D materials. We have now included the method to the introduction and to Table 2:

Introduction:

Four-point bending as an established adhesion measurement method^{50,51} was recently used to assess the adhesion of transition metal dichalcogenides (TMDC)²².

Table 2. Comparison of adhesion measurement methods of 2D materials. The button shear test method provides viable measurements in fracture mode II with low to moderate effort on rigid substrates like silicon.

measurement method	dominant fracture mode	lateral resolution	accuracy	sample preparation effort	measurement effort
blister test ^{16,17}	mixed mode	nm- μ m	high	high	high
nanoparticles ^{18,19}	mixed mode	nm- μ m	high	moderate	high
scratch testing ^{20,21}	mixed mode	μ m	moderate	moderate	low

Four-point bending ²²	mixed-mode	mm	moderate	moderate	low
nanoindentation ^{23,24}	mode I	nm	high	low	high
cantilever beam method ^{25,26}	mode I	mm	moderate	moderate	low
Micro force sensing during cleavage ¹⁵	mode II	μm	high	moderate	high
AFM friction measurement ²⁷	mode II	nm	low	low	high
substrate stretching on flexible substrates ^{5,28,29}	mode II	μm	moderate	low	moderate
button shear testing on rigid substrates ^{this work}	mode II	μm	moderate	moderate	low

5) Regarding to sample preparation, was Al film etched after PMMA etching? If so, it should be mentioned somewhere in page #66-70.

Thank you for pointing out this lack of clarity. We did not etch the Al film after PMMA etching. During process evaluation, we tried both: etching of Al film as the last processing step and keeping the Al film on top of the buttons. We did not detect any difference in delamination behavior. The higher optical contrast and faster identification of buttons with the Al film on top was the reason for keeping Al on top of the buttons.

In addition, by limiting the Al thickness to 20 nm, we were able to see through the thin Al film with a laser scanning microscope before PMMA and graphene structuring as shown in the Supplementary Fig. 3a and c.

We have added the following sentence to the manuscript for clarification:

“The buttons were tested both after the removal of the Al film as the last processing step and with the Al film remaining on top of the buttons. We observed no differences in the delamination behavior. The presence of Al provided a higher optical contrast and, hence, faster identification of buttons, so the process option with Al on top of the buttons was chosen for our experiments.”

Reviewer #3:

Recommendation: Publish as is; no revisions needed.

We thank the reviewer for this very positive feedback. We hope that the revisions we have made to the manuscript to answer the questions and concerns of the other two reviewers will keep reviewer #3 convinced of the merits of our work.

Bibliography of the point by point response letter:

1. Durix, L., Dreßler, M., Coutellier, D. & Wunderle, B. On the development of a modified button shear specimen to characterize the mixed mode delamination toughness. *Engineering Fracture Mechanics* **84**, 25–40 (2012).
2. Pufall, R. *et al.* Degradation of moulding compounds during highly accelerated stress tests – A simple approach to study adhesion by performing button shear tests. *Microelectronics Reliability* **52**, 1266–1271 (2012).
3. Ordonez, J. S., Boehler, C., Schuettler, M. & Stieglitz, T. Long-term Adhesion Studies of Polyimide to Inorganic and Metallic Layers. *MRS Proc.* **1466**, mrs12-1466-tt01-06 (2012).
4. Xu, C. *et al.* Rate-Dependent Decohesion Modes in Graphene-Sandwiched Interfaces. *Adv. Mater. Interfaces* **6**, 1901217 (2019).
5. Xu, C., Xue, T., Qiu, W. & Kang, Y. Size Effect of the Interfacial Mechanical Behavior of Graphene on a Stretchable Substrate. *ACS Appl. Mater. Interfaces* **8**, 27099–27106 (2016).
6. Shohji, I., Shimoyama, S., Ishikawa, H. & Kojima, M. Effect of Shear Speed on the Ball Shear Strength of Sn-3Ag-0.5Cu Solder Ball Joints. *Transactions of The Japan Institute of Electronics Packaging* **1**, 9–14 (2008).
7. Na, S. R. *et al.* Selective Mechanical Transfer of Graphene from Seed Copper Foil Using Rate Effects. *ACS Nano* **9**, 1325–1335 (2015).
8. Shohji, I., Yoshida, T., Takahashi, T. & Hioki, S. Tensile properties of Sn–Ag based lead-free solders and strain rate sensitivity. *Materials Science and Engineering: A* **366**, 50–55 (2004).
9. Feng, X. *et al.* Competing Fracture in Kinetically Controlled Transfer Printing. *Langmuir* **23**, 12555–12560 (2007).
10. Meitl, M. A. *et al.* Transfer printing by kinetic control of adhesion to an elastomeric stamp. *Nature Mater* **5**, 33–38 (2006).
11. Gent, A. N. Adhesion and Strength of Viscoelastic Solids. Is There a Relationship between Adhesion and Bulk Properties? *Langmuir* **12**, 4492–4496 (1996).
12. Christöfl, P. *et al.* Comprehensive investigation of the viscoelastic properties of PMMA by nanoindentation. *Polymer Testing* **93**, 106978 (2021).
13. Ionita, D., Cristea, M. & Banabic, D. Viscoelastic behavior of PMMA in relation to deformation mode. *J Therm Anal Calorim* **120**, 1775–1783 (2015).
14. Persson, B. N. J. & Brener, E. A. Crack propagation in viscoelastic solids. *Phys. Rev. E* **71**, 036123 (2005).
15. Wang, W. *et al.* Measurement of the cleavage energy of graphite. *Nat Commun* **6**, 7853 (2015).
16. Koenig, S. P., Boddeti, N. G., Dunn, M. L. & Bunch, J. S. Ultrastrong adhesion of graphene membranes. *Nat. Nanotechnology.* **6**, 543–546 (2011).
17. Calis, M., Lloyd, D., Boddeti, N. & Bunch, J. S. Adhesion of 2D MoS₂ to Graphite and Metal Substrates Measured by a Blister Test. *Nano Lett.* **23**, 2607–2614 (2023).
18. Zong, Z., Chen, C.-L., Dokmeci, M. R. & Wan, K. Direct measurement of graphene adhesion on silicon surface by intercalation of nanoparticles. *Journal of Applied Physics* **107**, 026104 (2010).

19. Dai, Z., Sanchez, D. A., Brennan, C. J. & Lu, N. Radial buckle delamination around 2D material tents. *J. Mech. Phys.* **137**, 103843 (2020).
20. Das, S., Lahiri, D., Lee, D.-Y., Agarwal, A. & Choi, W. Measurements of the adhesion energy of graphene to metallic substrates. *Carbon* **59**, 121–129 (2013).
21. Ivanov, E., Batakaliiev, T., Kotsilkova, R., Otto, M. & Neumaier, D. Study on the Adhesion Properties of Graphene and Hexagonal Boron Nitride Monolayers in Multilayered Micro-devices by Scratch Adhesion Test. *J. of Materi Eng and Perform* **30**, 5673–5681 (2021).
22. Nguyen, V. L. *et al.* Wafer-scale integration of transition metal dichalcogenide field-effect transistors using adhesion lithography. *Nature Electronics* **6**, 146–153 (2023).
23. Li, Y., Huang, S., Wei, C., Wu, C. & Mochalin, V. N. Adhesion of two-dimensional titanium carbides (MXenes) and graphene to silicon. *Nat Commun* **10**, 3014 (2019).
24. Rokni, H. & Lu, W. Direct measurements of interfacial adhesion in 2D materials and van der Waals heterostructures in ambient air. *Nat Commun* **11**, 5607 (2020).
25. Yoon, T. *et al.* Direct Measurement of Adhesion Energy of Monolayer Graphene As-Grown on Copper and Its Application to Renewable Transfer Process. *Nano Letters* **12**, 1448–1452 (2012).
26. Na, S. R., Suk, J. W., Ruoff, R. S., Huang, R. & Liechti, K. M. Ultra Long-Range Interactions between Large Area Graphene and Silicon. *ACS Nano* **8**, 11234–11242 (2014).
27. Zeng, X., Peng, Y. & Lang, H. A novel approach to decrease friction of graphene. *Carbon* **118**, 233–240 (2017).
28. Jiang, T., Huang, R. & Zhu, Y. Interfacial Sliding and Buckling of Monolayer Graphene on a Stretchable Substrate. *Adv. Funct. Mater.* **24**, 396–402 (2014).
29. Wang, G. *et al.* Tuning the Interfacial Mechanical Behaviors of Monolayer Graphene/PMMA Nanocomposites. *ACS Appl. Mater. Interfaces* **8**, 22554–22562 (2016).
30. Dai, Z. *et al.* Mechanical behavior and properties of hydrogen bonded graphene/polymer nano-interfaces. *Compos. Sci. Technol.* **136**, 1–9 (2016).
31. Li, R. *et al.* Determination of PMMA Residues on a Chemical-Vapor-Deposited Monolayer of Graphene by Neutron Reflection and Atomic Force Microscopy. *Langmuir* **34**, 1827–1833 (2018).
32. Choi, W., Shehzad, M. A., Park, S. & Seo, Y. Influence of removing PMMA residues on surface of CVD graphene using a contact-mode atomic force microscope. *RSC Adv.* **7**, 6943–6949 (2017).
33. Zhuang, B., Li, S., Li, S. & Yin, J. Ways to eliminate PMMA residues on graphene — superclean graphene. *Carbon* **173**, 609–636 (2021).
34. Ahn, Y., Kim, J., Ganorkar, S., Kim, Y.-H. & Kim, S.-I. Thermal annealing of graphene to remove polymer residues. *Mat Express* **6**, 69–76 (2016).
35. Zheng, F. *et al.* Critical Stable Length in Wrinkles of Two-Dimensional Materials. *ACS Nano* **14**, 2137–2144 (2020).
36. Aitken, Z. H. & Huang, R. Effects of mismatch strain and substrate surface corrugation on morphology of supported monolayer graphene. *Journal of Applied Physics* **107**, 123531 (2010).

37. Li, T. & Zhang, Z. Substrate-regulated morphology of graphene. *J. Phys. D: Appl. Phys.* **43**, 075303 (2010).
38. Gao, W. & Huang, R. Effect of surface roughness on adhesion of graphene membranes. *J. Phys. D: Appl. Phys.* **44**, 452001 (2011).
39. Spear, J. C., Custer, J. P. & Batteas, J. D. The influence of nanoscale roughness and substrate chemistry on the frictional properties of single and few layer graphene. *Nanoscale* **7**, 10021–10029 (2015).
40. Deng, S., Gao, E., Xu, Z. & Berry, V. Adhesion Energy of MoS₂ Thin Films on Silicon-Based Substrates Determined via the Attributes of a Single MoS₂ Wrinkle. *ACS Appl. Mater. Interfaces* **9**, 7812–7818 (2017).
41. Pflügler, N. *et al.* Experimental determination of critical adhesion energies with the Advanced Button Shear Test. *Microelectronics Reliability* **99**, 177–185 (2019).
42. Sanchez, D. A. *et al.* Mechanics of spontaneously formed nanoblister traps by transferred 2D crystals. *Proc Natl Acad Sci USA* **115**, 7884–7889 (2018).
43. Torres, J., Zhu, Y., Liu, P., Lim, S. C. & Yun, M. Adhesion Energies of 2D Graphene and MoS₂ to Silicon and Metal Substrates. *Phys. Status Solidi A* **215**, 1700512 (2018).
44. Li, Y. *et al.* Adhesion Between MXenes and Other 2D Materials. *ACS Appl. Mater. Interfaces* **13**, 4682–4691 (2021).
45. Lloyd, D. *et al.* Adhesion, Stiffness, and Instability in Atomically Thin MoS₂ Bubbles. *Nano Lett.* **17**, 5329–5334 (2017).
46. Wagner, S. *et al.* Graphene transfer methods for the fabrication of membrane-based NEMS devices. *Microelectronic Engineering* **159**, 108–113 (2016).
47. Marx, M. *et al.* Metalorganic Vapor-Phase Epitaxy Growth Parameters for Two-Dimensional MoS₂. *Journal of Elec Materi* **47**, 910–916 (2018).
48. Cun, H. *et al.* Wafer-scale MOCVD growth of monolayer MoS₂ on sapphire and SiO₂. *Nano Res.* **12**, 2646–2652 (2019).
49. Grundmann, A. *et al.* Impact of synthesis temperature and precursor ratio on the crystal quality of MOCVD WSe₂ monolayers. *Nanotechnology* **34**, 205602 (2023).
50. Dauskardt, R. H., Lane, M., Ma, Q. & Krishna, N. Adhesion and debonding of multi-layer thin film structures. *Engineering Fracture Mechanics* **61**, 141–162 (1998).
51. Birringer, R. P., Chidester, P. J. & Dauskardt, R. H. High yield four-point bend thin film adhesion testing techniques. *Engineering Fracture Mechanics* **78**, 2390–2398 (2011).

REVIEWERS' COMMENTS

Reviewer #1 (Remarks to the Author):

The authors have addressed most of my concerns; however, I believe that further modifications are still needed for this work to be published in Nature Communications.

In my understanding, the button shear technique refers to any prefabricated feature that can be utilized to shear one material with respect to the other for the examination of interlayer interaction characteristics (e.g., Nature Communications, volume 14, Article number: 2931 (2023)). In this light, the contribution of this work is to use a polymer button to study the shear strength of CVD-grown monolayer 2D crystals on the underlying SiO₂ and Si₃N₄ substrates. To further improve the proposed technique, it would be beneficial to address the main drawbacks of not using a “rigid polymer” button.

To further demonstrate the capability of this technique and quantify the influence of defect densities in CVD-grown 2D crystals on the reported shear strength, I still recommend measuring the critical shear strength between mechanically exfoliated monolayer 2D crystals and the underlying substrates. In response to the authors' concern regarding the small lateral size of mechanically exfoliated 2D crystals, it is worth noting that large-area mechanical exfoliation of single-crystal 2D materials up to a millimeter in size has been reported in the literature (e.g., Nature Communications volume 11, Article number: 2453 (2020); SCIENCE ADVANCES, 28 Oct 2020, Vol 6, Issue 44; ACS Nano 2015, 9, 11, 10612–10620).

Reviewer #2 (Remarks to the Author):

The authors have fully addressed my comments and concerns. I would be happy to recommend the publication of this work.

Point by point response letter

For the manuscript "Button Shear Testing for Adhesion Measurements of 2D Materials"

Reviewer #1:

The authors have addressed most of my concerns; however, I believe that further modifications are still needed for this work to be published in Nature Communications.

- 1) *In my understanding, the button shear technique refers to any prefabricated feature that can be utilized to shear one material with respect to the other for the examination of interlayer interaction characteristics (e.g., Nature Communications, volume 14, Article number: 2931 (2023)). In this light, the contribution of this work is to use a polymer button to study the shear strength of CVD-grown monolayer 2D crystals on the underlying SiO₂ and Si₃N₄ substrates. To further improve the proposed technique, it would be beneficial to address the main drawbacks of not using a "rigid polymer" button.*

We agree with the reviewer's view on the contribution of this work: "... to use a polymer button to study the shear strength of CVD-grown monolayer 2D crystals on the underlying SiO₂ and Si₃N₄ substrates." Depending on the focus it is possible to focus on two aspects of this sentence:

- a) From an electronic systems point of view, it is an important step to widely applicable adhesion measurements of 2D materials. The existing shear testing tools for e.g. mold compound adhesion measurements^{1,2} and knowledge thereof can be applied to assess the adhesion of this material group now.
- b) From a 2D material community point of view, it is the extension of different shearing approaches by using a polymer as an assistance layer instead of other thin films. One advantage is the application of commercial shear testers like Dage4000Plus that guarantee reproducible and comparable results of shear strength by different research groups. These tools are widespread and comply with various industry standards. On the other hand, the viscoelasticity (discussed previously) and the lower hardness are challenges compared to other assistance layers.

In the original manuscript we highlighted the electronic systems view a). Reviewer #1 challenges aspects of view b). We are happy about that as this leads to a more comprehensive manuscript.

The low elastic modulus leads to a button deformation near the contacted button edge and to a pronounced mixed mode of forces perpendicular (mode I) and parallel (mode II) to the surface near the contacted button edge. To our understanding, this is by far the largest effect of the low elastic modulus of PMMA in button shear testing. Therefore, we embedded the drawback-section of PMMA into the discussion on mixed mode:

A button length of 60 μm was established as a minimum for reproducible results by these initial experiments because smaller buttons with 20 μm and 40 μm button length show very low and scattered F_c (Error! Reference source not found.d). **PMMA is a soft material with low elastic modulus compared to other assistance layers in shear testing experiments with 2D materials³. That leads to button deformation near the contacted button edge. Hence,** the ratio of forces perpendicular (mode I) and parallel (mode II) to the surface varies along the shear path^{2,4}. The pronounced mixed mode near the **contacted** edge can be the reason for the low F_c at small button lengths⁵.

Furthermore, we combine these considerations with observations made in Supplementary Figs. 3-5: 2D material residues are found predominantly near the contacted button edge.

We added in the manuscript:

In all cases, delamination at the substrate-graphene interface occurred. This was confirmed by optical microscopy (Supplementary Fig. 4) and Raman measurements (Supplementary Fig. 5). **Minor graphene residues are found predominantly near the contacted button edge. The pronounced mixed mode at the contacted button edge is a likely reason for this observation⁵.**

Finally, we challenge the reproducibility of the PMMA button fabrication. Changes in PMMA button fabrication may lead to changes in PMMA properties and shear strength results. We achieved similar results for the critical shear force of PMMA on thermal SiO₂ even after several months within our last experiments. Therefore, we conclude that it is possible to control the PMMA button fabrication and achieve reproducible results.

We added in the manuscript:

Control of the mechanical properties of the button material is crucial for reproducible results. As discussed previously, the relatively soft PMMA leads to button deformation and pronounced mixed mode near the contacted button edge. Variations in button fabrication can lead to changes in the PMMA properties and, hence, to changes in the measured shear strength⁵. The shear strength of PMMA on thermal SiO₂ is $\tau_c = 31.84 \pm 2.61$ MPa on the MoS₂ sample and $\tau_c = 30,00 \pm 7,18$ MPa on the WSe₂ sample. Previous measurements of PMMA on thermal SiO₂ in **Error! Reference source not found.d** with 60 μm buttons ($F_c = 189.75 \pm 1.99$ mN, $\tau_c = 31.63 \pm 0.33$ MPa) and in **Error! Reference source not found.c** with 10 μm s⁻¹ ($\tau_c = 34.60$ MPa) performed months before these measurements led to similar τ_c , confirming the reproducibility of button shear testing **with PMMA as button material.**

- 2) *To further demonstrate the capability of this technique and quantify the influence of defect densities in CVD-grown 2D crystals on the reported shear strength, I still recommend measuring the critical shear strength between mechanically exfoliated monolayer 2D crystals and the underlying substrates. In response to the authors' concern regarding the small lateral size of mechanically exfoliated 2D crystals, it is worth noting that large-area mechanical exfoliation of single-crystal 2D materials up to a millimeter in size has been reported in the literature (e.g., Nature Communications volume 11, Article number: 2453 (2020); SCIENCE ADVANCES, 28 Oct 2020, Vol 6, Issue 44; ACS Nano 2015, 9, 11, 10612–10620).*

We still feel unable to fulfil this request. Fig. 5a,b displays the typical lateral size of our 2D material (10 by 10 μm). Exfoliated material with dimensions of 1 by 1 mm or 2 by 2 mm will not lead to sufficient statistics. Furthermore, in this aspect, we remain on the electronic systems point of view that is interested in CVD-grown material instead of exfoliated crystals.

Reviewer #2:

The authors have fully addressed my comments and concerns. I would be happy to recommend the publication of this work.

We are happy that our revisions satisfy reviewer #2 and thank the reviewer for dedicating his time to review our revised manuscript.

Bibliography of the point by point response letter:

1. Goroll, M. & Pufall, R. New aspects in characterization of adhesion of moulding compounds on different surfaces by using a simple button-shear-test method for lifetime prediction of power devices. *Microelectronics Reliability* **50**, 1684–1687 (2010).
2. Pufall, R. *et al.* Degradation of moulding compounds during highly accelerated stress tests – A simple approach to study adhesion by performing button shear tests. *Microelectronics Reliability* **52**, 1266–1271 (2012).
3. Huang, X. *et al.* Robust microscale structural superlubricity between graphite and nanostructured surface. *Nat Commun* **14**, 2931 (2023).
4. Dudek, R. *et al.* Determination of interface fracture parameters by shear testing using different theoretical approaches. in *2012 13th International Thermal, Mechanical and Multi-Physics Simulation and Experiments in Microelectronics and Microsystems* 1/10-10/10 (IEEE, Cascais, Portugal, 2012). doi:10.1109/ESimE.2012.6191793.
5. Durix, L., Dreßler, M., Coutellier, D. & Wunderle, B. On the development of a modified button shear specimen to characterize the mixed mode delamination toughness. *Engineering Fracture Mechanics* **84**, 25–40 (2012).